# A Unified Approach to Routing and Cascading for LLMs

## Abstract

The widespread applicability of large language models (LLMs) has increased the availability of many fine-tuned models of various sizes targeting specific tasks. Given a set of such specialized models, to maximize overall performance, it is important to figure out the optimal strategy for selecting the right model for a given user query. An effective strategy could drastically increase overall performance and even offer improvements over a single large monolithic model. Existing approaches typically fall into two categories: routing, where a single model is selected for each query, and cascading, which runs a sequence of increasingly larger models until a satisfactory answer is obtained. However, both have notable limitations: routing commits to an initial model without flexibility, while cascading requires executing every model in sequence, which can be inefficient. Additionally, the conditions under which these strategies are provably optimal remain unclear. In this work, we derive optimal strategies for both routing and cascading. Building on this analysis, we propose a novel approach called *cascade routing*, which combines the adaptability of routing with the cost-efficiency of cascading. Our experiments demonstrate that cascade routing consistently outperforms both routing and cascading across a variety of settings, improving both output quality and lowering computational cost, thus offering a unified and efficient solution to the model selection problem.[1]

## 1 Introduction

Large language models (LLMs) have found applications in a wide range of tasks, some of which are easily handled by small models, while others require the full capacity of state-of-the-art LLMs. This has led to the development of many fine-tuned models of various sizes that target specific tasks. To maximize performance, it is crucial to select the most suitable model for each query, accounting for both the expected quality of the model's output and the model's cost. Such model selection strategies can significantly improve performance over any individual model and can reduce inference costs by selecting a smaller model when the query does not require the full capacity of a larger model.

**Routing and Cascading** Two primary strategies have been proposed to solve model selection. The first, routing, directs each input query to a specific model from a set of available models (Chen et al., 2022; Liu et al., 2024), as illustrated in Fig. 1(a). This approach is particularly effective when different expert LLMs are needed for different tasks, enabling the selection of the most suitable expert for each query. The second strategy, cascading, processes an input query through a sequence of increasingly larger models, stopping when a model produces an answer deemed sufficiently good (Chen et al., 2023; Varshney and Baral, 2022), as illustrated in Fig. 1(b). Cascading is particularly valuable for handling queries of varying difficulty, as it allows simpler queries to be addressed by smaller, less expensive models while reserving more complex queries for larger models.

**Restrictive Conditions** Despite their utility, both routing and cascading impose significant restrictions on the model selection process. In routing, the initial selection of a model is final, preventing any reconsideration after the initial decision. In cascading, each query must sequentially pass through all models in the chain until a suitable answer is found, with no option to reroute to a potentially better model. Therefore, a less restrictive strategy that combines the strengths of both routing and cascading could offer significant performance improvements.

---

[1]Code available in supplementary material.

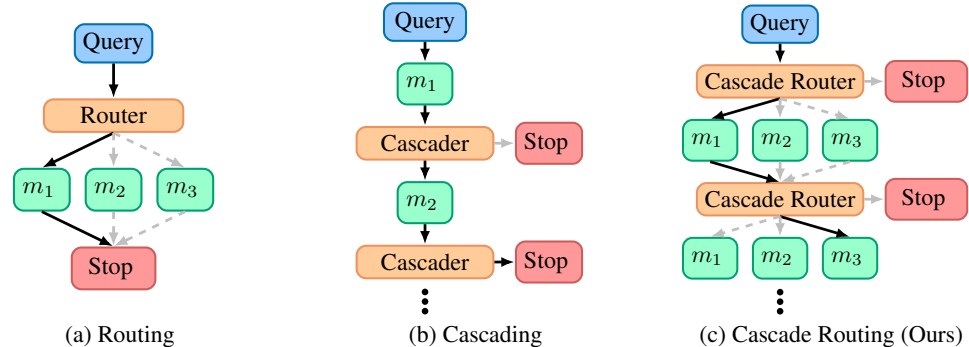

(a) Routing        (b) Cascading        (c) Cascade Routing (Ours)

Figure 1: Overview of three model selection strategies. Routing selects a single model for a query, cascading processes queries through a sequence of models, and cascade routing generalizes both.

**Lack of Theoretical Understanding** Furthermore, the conditions under which current routing and cascading strategies are optimal, are not well understood. For routing, an extensive proof is required just to show that current strategies are *close to* optimal (Chen et al., 2022), while the theoretical analysis of current cascading algorithms is based on restrictive assumptions and does not provide optimality guarantees (Chen et al., 2023; Varshney and Baral, 2022). This lack of theoretical understanding hinders the development of more effective model selection strategies.

**This Work: Cascade Routing** To address these limitations, we first derive optimal routing and cascading strategies by framing them as linear optimization problems aimed at maximizing output quality while remaining within a given cost budget. For routing, this optimal strategy is close to the one obtained by prior work, while for cascading we derive a new strategy that is provably better than existing approaches. Building on this theoretical analysis, we propose a new paradigm called cascade routing, which generalizes both routing and cascading. As illustrated in Fig. 1(c), cascade routing initially routes a query to any available model but keeps rerouting to different models until a model produces an answer of sufficient quality. We prove the optimality of our cascade routing strategy and show that it offers significantly more flexibility in processing a query.

**Results** We evaluate cascade routing on a wide range of tasks, demonstrating that it significantly outperforms both routing and cascading. Notably, cascade routing consistently improves performance by $1\%$ to $4\%$ across all settings on the popular RouterBench benchmark (Hu et al., 2024), which represents a relative improvement over a naive baseline by an additional $13\%$ to $80\%$. Furthermore, we show that our new cascading strategy outperforms existing cascading strategies by up to $2\%$, validating that our theoretical analysis leads to practical improvements over prior work.

**Contributions** Our main contributions are as follows:

- We derive optimal strategies for routing and cascading and obtain a new cascading strategy that is provably better than prior approaches (§2, §3).
- We introduce cascade routing, a new paradigm that combines the strengths of routing and cascading, and prove its optimality (§4).
- We conduct a thorough evaluation, demonstrating that cascade routing consistently outperforms the baselines (§5).

## 2 ROUTING AS LINEAR OPTIMIZATION

We derive an optimal routing strategy to select the best model for a given query, providing detailed proofs for all statements in this section in App. A. We will use the analysis presented here to develop the optimal cascading and cascade routing strategies in §3 and §4, respectively.

**Routing Strategy** In routing, our goal is to develop a strategy that selects the best language model for a given input query. Formally, let $\mathcal{X}$ represent the distribution over all possible queries, and suppose we have $k$ language models $m_1, \ldots, m_k$ available for routing. Further, let $\Delta_k$ denote the set of all probability distributions over $k$ variables. A routing strategy can then be defined as follows:

**Definition 1** (Routing). *A routing strategy $s$ is a function $s\colon \mathcal{X} \to \Delta_k$ that maps a query $x \in \mathcal{X}$ to a probability distribution over models. $s_i(x)$ denotes the probability that $m_i$ is selected for query $x$.*

A routing strategy selects a model by sampling from the distribution $s(x)$ for each query $x$. In prior work, routing strategies were restricted to be deterministic, i.e., $s_i(x) \in \{0, 1\}$ (Chen et al., 2022; Hu et al., 2024). In contrast, we propose using a more general probabilistic routing strategy that enables a better solution and an easier theoretical analysis.

**Quality and Cost**   In routing, we seek to maximize the expected output quality of the selected model while adhering to a given cost budget $B$. Quality could measure model accuracy, user preference, or any other performance indicator. We define the quality function $q_i(x)$ as the output quality of model $m_i$ on query $x$, and the cost function $c_i(x)$ as the cost of running model $m_i$ on $x$.

However, since these functions are unknown in practice, we need estimators $\hat{q}_i(x)$ and $\hat{c}_i(x)$ that approximate the output quality and cost of querying model $m_i$ on input $x$. Estimators for $q_i(x)$ can be created using small classifiers trained to predict model accuracy, as done in prior work (Hu et al., 2024; Shnitzer et al., 2023). $\hat{c}_i(x)$ can be estimated by tokenizing the input query and determining the average output length of the model on a query. Then, we can use API-specific costs per token to estimate the cost of running a model on a query.

**Optimal Routing**   Using these estimators, we can formally define the optimal routing strategy:

**Definition 2** (Optimal Routing). *The optimal routing strategy $s_{\mathrm{OPT}}$ for a given cost budget $B$ is the solution to the optimization problem*

$$\max_s \quad \mathbb{E}_{x \in \mathcal{X}, i \in \{1,...,k\}}(s_i(x)\hat{q}_i(x))$$
$$s.t. \quad \mathbb{E}_{x \in \mathcal{X}, i \in \{1,...,k\}}(s_i(x)\hat{c}_i(x)) \leqslant B. \tag{1}$$

We now explain how to solve this linear optimization problem. Intuitively, the optimal routing strategy optimizes the cost-quality tradeoff $\tau_i(x, \lambda) = \hat{q}_i(x) - \lambda\hat{c}_i(x)$, where $\lambda \in \mathbb{R}^+$ controls the balance between quality and cost based on the budget $B$. For each query $x$, the model that achieves the highest value of $\tau_i(x, \lambda)$ is selected.

More formally, for a given $\lambda \in \mathbb{R}^+$, we define two deterministic routing strategies: $s_{\mathrm{MIN}}^\lambda(x)$, which selects the cheapest model achieving the optimal cost-quality tradeoff, and $s_{\mathrm{MAX}}^\lambda(x)$, which selects the most expensive model achieving this tradeoff. The optimal routing strategy $s_{\mathrm{OPT}}$ is then determined by the following theorem:

**Theorem 1** (Optimal Routing Strategy). *For a given cost budget $B$, there exists a $\lambda \in \mathbb{R}^+$ and a $\gamma \in [0, 1]$ such that the optimal routing strategy $s_{\mathrm{OPT}}$ equals $\gamma s_{\mathrm{MIN}}^\lambda + (1 - \gamma)s_{\mathrm{MAX}}^\lambda$. Furthermore, all routing strategies that have an expected cost that is exactly equal to $B$ and can be written as a convex combination of $s_{\mathrm{MIN}}^{\lambda'}$ and $s_{\mathrm{MAX}}^{\lambda'}$ for some $\lambda' \in \mathbb{R}^+$ achieve the same optimal quality.*

Since $\gamma$ is often not equal to 0 or 1, the optimal routing strategy is not deterministic and instead selects a model probabilistically. Therefore, prior work that only considered deterministic routing strategies (Chen et al., 2022; Hu et al., 2024) cannot express the routing strategy from Theorem 1 and fall back to the near-optimal $s_{\mathrm{MIN}}^\lambda$ instead.

Due to the second part of Theorem 1, we only need to find *a* set of hyperparameters $\lambda$ and $\gamma$ that achieve the desired cost budget. To determine these parameters, we estimate the cost of each strategy using a validation dataset $D$ that is representative of the query distribution $\mathcal{X}$. Since the cost of $s_{\mathrm{MIN}}^\lambda$ increases as $\lambda$ decreases (see App. A), we use a binary search to find the appropriate value of $\lambda$. $\gamma$ is determined by interpolating between the costs of $s_{\mathrm{MIN}}^\lambda$ and $s_{\mathrm{MAX}}^\lambda$ to match the budget $B$.

## 3   CASCADING AS SEQUENTIAL ROUTING

In this section, we extend our analysis of the optimal routing strategy to the cascade setting, providing detailed proofs for all the statements in this section in App. B. The solution derived here will be used to develop the optimal strategy for cascade routing in §4.

**Cascading**   In cascading, an input query is passed through a chain of increasingly larger and more expensive models. The cascade stops once a model's output meets a certain condition, and that output is returned. We will reinterpret cascading as a sequence of routing problems. To do so, we first define the models over which we need to route, which we refer to as supermodels.

**Definition 3** (Supermodel). *A supermodel $M$ is a sequence of models $(m_{i_1}, \ldots, m_{i_j})$ such that running a query through $M$ is equivalent to running it through each of the models in the sequence. The set of all supermodels is denoted as $\mathcal{M}$. By $M_{i:j}$ we denote the supermodel $(m_i, \ldots, m_j)$.*

In cascading, we only need to consider the supermodels $M_{1:1}, \ldots, M_{1:k}$. The full expressivity of Definition 3 will only be necessary for cascade routing in §4.

A cascade occurs as a sequence of decision steps, where at each step, it decides whether to compute the next model in the sequence or terminate. By step $j$, the first $j-1$ models have been computed. At this point, the cascade will run one of the supermodels $M_{1:j-1}, \ldots, M_{1:k}$. If any of the supermodels $M_{1:j}, \ldots, M_{1:k}$ is optimal, the cascade proceeds to run the next model. In contrast, if the supermodel $M_{1:j-1}$ is optimal, the cascade halts and returns the output of the last computed model.

Thus, a cascade can be characterized as a sequence of routing strategies that route between the supermodels $M_{1:j-1}, \ldots, M_{1:k}$. Even though the action associated with supermodels $M_{1:j}, \ldots, M_{1:k}$ – continuing the cascade – is the same, it is essential to use all of them in the routing strategy. For instance, if $m_j$ performs poorly but $m_{j+1}$ performs exceptionally well on a given query, the cascade should continue. Limiting consideration to the supermodels $M_{1:j-1}$ and $M_{1:j}$ alone would therefore result in suboptimal decisions. Formally, we define a cascading strategy as follows:

**Definition 4** (Cascading Strategy). *A cascading strategy $s$ is a sequence of routing strategy $(s^{(1)}, \ldots, s^{(k)})$ such that $s^{(j)}$ routes between the supermodels $M_{1:j-1}, \ldots, M_{1:k}$.*

**Quality and Cost**   To apply Theorem 1 to find the optimal cascading strategy, we first need to derive the quality and cost estimates of the supermodels. Both of these can depend on the answers of previously computed models. Therefore, let $\hat{q}^{(j)}(x)$ and $\hat{c}^{(j)}(x)$ represent the updated estimates in step $j$ after computing the first $j - 1$ models.

We derive the quality and cost estimates associated with supermodel $M_{1:i}$, denoted as $\hat{q}_{1:i}^{(j)}(x)$ and $\hat{c}_{1:i}^{(j)}(x)$, based on the quality and cost estimates of the individual models. Trivially, the cost estimate of the supermodel is equal to the sum of the individual model costs. The quality of a supermodel, however, is governed by the best model within it. Thus, it equals $\mathbb{E}[\max(\hat{q}_1(x), \ldots, \hat{q}_i(x))]$, where the expected value reflects the uncertainty in each quality estimate. This expected value is crucial since ignoring uncertainty would falsely assume that the quality of a supermodel is always equal to the best model within it, even though the best model may return a poor answer, while another returns a good one. To estimate the uncertainties associated with the quality estimates, we compute the variance of $\hat{q}_i^{(j)}(x) - \hat{q}_i^{(k)}(x)$ over a validation dataset $D$.

While the form of these estimates works well in practice and is intuitively clear, we note that alternative approaches are possible. However, the optimality proof of our cascading strategy does not depend on the specific form of the estimates, and can therefore be adapted to other formulations.

**Optimal Cascading**   We now leverage the optimal routing strategy from Theorem 1 to determine the optimal cascading strategy. As before, optimality is defined in terms of maximizing the expected output quality while adhering to a given cost budget. However, the budget is now only enforced over the entire cascade, and not over the individual steps. This leads to a slightly different formulation of the optimal cascading strategy:

**Theorem 2** (Optimal Cascading Strategy). *For a given cost budget $B$, there exist $\lambda_1, \ldots, \lambda_k \in \mathbb{R}^+$ and a $\gamma \in [0, 1]$ such that the optimal cascading strategy $s_{\mathrm{OPT}} = (s_{\mathrm{OPT}}^{(1)}, \ldots, s_{\mathrm{OPT}}^{(k)})$ is given by the equalities $s_{\mathrm{OPT}}^{(j)} = \gamma s_{\mathrm{MIN}}^{(j), \lambda_j} + (1 - \gamma) s_{\mathrm{MAX}}^{(j), \lambda_j}$ where $s_{\mathrm{MIN}}^{(j), \lambda_j}$ and $s_{\mathrm{MAX}}^{(j), \lambda_j}$ are defined as in Theorem 1.*

The main difference between Theorem 2 and Theorem 1 is that not all combinations of hyperparameters $\lambda_1, \ldots, \lambda_k \in \mathbb{R}^+$ and $\gamma \in [0, 1]$ that achieve cost budget $B$ are optimal. Instead, finding the optimal hyperparameters requires solving another optimization problem. Specifically, let us denote by $Q_D \colon (\mathbb{R}^+)^{k+1} \to \mathbb{R}$, resp. $C_D \colon (\mathbb{R}^+)^{k+1} \to \mathbb{R}$, the average quality, resp. cost, of a cascading strategy on a training dataset $D$ for a given set of hyperparameters $\lambda_1, \ldots, \lambda_k \in \mathbb{R}^+$ and $\gamma \in [0, 1]$. Then, the optimal hyperparameters are the solution to the optimization problem

$$\max_{\lambda_1, \ldots, \lambda_k, \gamma} \quad Q_D(\lambda_1, \ldots, \lambda_k, \gamma)$$
$$\text{s.t.} \quad C_D(\lambda_1, \ldots, \lambda_k, \gamma) \leqslant B. \tag{2}$$

We could not find an analytical solution to this problem. Therefore, we assume $\lambda_1 = \cdots = \lambda_k = \lambda$ and apply the binary search technique from §2 to determine the optimal $\lambda$. This value is then used as initialization for a hyperparameter optimization tool, Hyperopt[2], to find the optimal values.

**Prior Work**   Prior work on cascading has often relied on strong assumptions to simplify the strategy. The most common technique uses a threshold to decide whether to continue the cascade on an input $x$ (Chen et al., 2023; Gupta et al., 2024). Specifically, in step $j$, the cascade continues if $\hat{q}_{j-1}^{(j)}(x) < \tau_j$ for some threshold $\tau_j \in \mathbb{R}$. Below, we outline the conditions under which this simplified approach is optimal.

**Corollary 1** (Optimal Threshold Strategy). *Under minor technical assumptions, the thresholding strategy is equivalent to our cascading strategy if and only if the following conditions hold: the cost estimate $\hat{c}_i^{(j)}(x)$ is independent of $x$ for all $i, j \in \{1, \ldots, k\}$, $\hat{q}_i^{(j)}(x)$ is independent of $x$ for all $i \geqslant j$, and the quality estimate $\hat{q}_{1:i}^{(j)}(x)$ is equal to $\hat{q}_i^{(j)}(x)$.*

# 4 CASCADE ROUTING AS CASCADE GENERALIZATION

Both routing and cascading are powerful techniques that enable the efficient use of multiple models. However, their use is often orthogonal: while routing is useful when we have several specialized models that are experts at specific tasks, cascading is more beneficial when input queries vary in difficulty. In this section, we therefore present cascade routing, which is a generalization of both techniques. Proofs for all theorems and lemmas in this section are included in App. C.

**Cascade Routing**   Cascade routing closely resembles cascading, but with one crucial difference: the routing strategy at step $j$ routes between all possible supermodels, not just the supermodels $M_{1:j-1}, \ldots, M_{1:k}$. Therefore, both Definition 4 and Theorem 2 can be extended to this setting.

**Definition 5** (Cascade Routing). *A cascade routing strategy $s$ is a sequence of routing strategies $(s^{(1)}, \ldots, s^{(k)})$ such that, for a given sample $x \in \mathcal{X}$, $s^{(j)}$ routes between all supermodels in $\mathcal{M}$ that start with the $j-1$ models that have already been computed for this query.*

**Theorem 3** (Optimal Cascade Routing). *For a given cost budget $B$, there exist $\lambda_1, \ldots, \lambda_k \in \mathbb{R}^+$ and a $\gamma \in \mathbb{R}^+$ such that the optimal cascade routing strategy $s_{\text{OPT}} = (s_{\text{OPT}}^{(1)}, \ldots, s_{\text{OPT}}^{(k)})$ is given by $s_{\text{OPT}}^{(j)} = \gamma s_{\text{MIN}}^{(j), \lambda_j} + (1 - \gamma) s_{\text{MAX}}^{(j), \lambda_j}$ where $s_{\text{MIN}}^{(j), \lambda_j}$ and $s_{\text{MAX}}^{(j), \lambda_j}$ are defined as in Theorem 1.*

While cascade routing extends cascading and can therefore use the same hyperparameter optimization scheme, it also introduces additional challenges which we address in the following paragraphs.

**Model Order**   In cascading, the model order is predetermined, and the routing strategy only decides whether to proceed with the next model in the sequence. In contrast, cascade routing must dynamically determine the order in which models are computed. Despite this, both the estimated quality $\hat{q}_M^{(j)}(x)$ and cost $\hat{c}_M^{(j)}(x)$ of a supermodel $M$ are order-independent. Therefore, supermodels that contain the same models in a different order will have the same associated cost and quality. To mitigate this, we sort the models within the selected supermodel by cost and compute the cheapest one first. This approach aligns with cascading, where more expensive models are only used if cheaper models do not suffice.

**Number of Supermodels**   In cascading, the quality and cost must be computed for a maximum of $k$ supermodels at each step. However, in cascade routing, the number of supermodels grows exponentially, leading to the need to evaluate up to $2^k$ supermodels. This increase can become prohibitively costly, particularly since the model selection process must remain computationally negligible with respect to model computation. To mitigate this, we leverage so-called negative marginal gains. Specifically, if a model $m$ in a supermodel $M$ negatively impacts the quality-cost tradeoff, all supermodels containing all models in $M$ can be pruned from the search space. Since this negative contribution is quite common, this allows us to prune the search space significantly. More formally, this pruning operation relies on the following lemma:

**Lemma 1** (Negative Marginal Gain). *Let $M \in \mathcal{M}$ and $m$ be any model in $M$. Let the marginal gain of $m$ w.r.t. $M$ be defined as $\tau_M(x, \lambda) - \tau_{M \setminus \{m\}}(x, \lambda)$. Then, if the marginal gain of $m$ w.r.t. $M$ is strictly negative for a given query, the optimal cascade routing strategy will never run a supermodel $M' \in \mathcal{M}$ that contains all models in $M$.*

---

[2]https://github.com/hyperopt/hyperopt

## 5 EVALUATION

We evaluate the performance of cascade routing and demonstrate that it significantly outperforms all other strategies. Additionally, we show that our new cascading approach surpasses the threshold-based cascading method (see Corollary 1). For this purpose, we first conduct experiments on Router-Bench (Hu et al., 2024), a benchmark specifically designed to evaluate routing and cascading (§5.1). Next, we test cascade routing on two additional benchmarks to evaluate its performance in more realistic scenarios (§5.2). Lastly, we perform an ablation study to examine the impact of various design choices in cascade routing on performance and runtime (§5.3).

### 5.1 ROUTERBENCH

RouterBench (Hu et al., 2024) is a benchmark developed to evaluate the efficacy of different model selection strategies. It includes questions from seven diverse benchmarks, such as MMLU (Hendrycks et al., 2021), GSM8k (Cobbe et al., 2021), and MBPP (Austin et al., 2021), alongside answers from eleven different models ranging from GPT-4 (OpenAI, 2023) to Mistral-7B (Jiang et al., 2023). All models are evaluated under both zero-shot and five-shot settings.

**Quality and Cost Estimates**  Similar to (Hu et al., 2024), we estimate quality and cost by adding zero-centered Gaussian noise to their true values. Both cost and quality estimates are modeled as linear functions fitted on these noisy signals. We define the variance of the noisy signal as $\sigma^2_{\text{before}}$ before model computation and $\sigma^2_{\text{after}}$ after. By ensuring that $\sigma_{\text{after}} < \sigma_{\text{before}}$, this setup reflects the increased accuracy in cost and quality estimates after model computation, which is an essential requirement for cascading to perform well. To explore different uncertainty levels, we vary the variances to simulate low-, medium-, and high-noise scenarios, with exact values for the variances detailed in App. D.

**Models**  We evaluate cascade routing on RouterBench using three, five, and all eleven models available for model selection, ensuring a comprehensive evaluation across a range of scenarios. The exact models for each scenario are provided in App. D.

**Strategies**  We compare cascade routing against several baseline strategies, including the routing strategy described in §2, the threshold-based cascading approach from prior work (Corollary 1), and the optimal cascading strategy (Theorem 2). Additionally, as in (Hu et al., 2024), we include a baseline that linearly interpolates cost and quality on the Pareto frontier of the models. Fig. 2 illustrates the performance of these strategies with five models in a medium-noise setting.

**Evaluation Metric**  For each method, we evaluate performance using cost budgets ranging from the cheapest to the most expensive model, as shown in Fig. 2. This produces a quality-cost curve for each strategy. Following (Hu et al., 2024), we use the Area Under the Curve (AUC) as the performance metric.

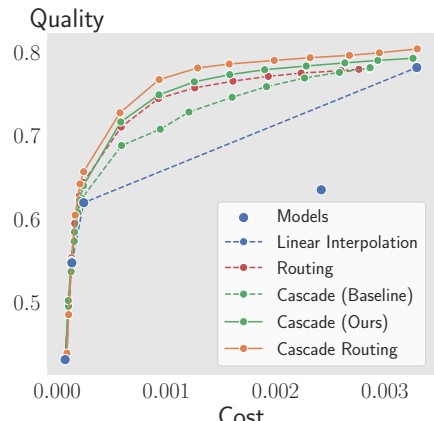

Figure 2: Quality-cost tradeoff on Router-Bench for five models and medium-noise.

**Results**  Table 1 presents the results for the zero-shot setting, with the five-shot results detailed in App. I. Cascade routing consistently outperforms all baseline strategies with performance gains between 1% to 4%, which measured relatively to the naive linear interpolation baseline means that cascade routing improves by 13% to 80% over the baselines. This performance gap widens as more models are available and narrows under higher noise levels, indicating that cascade routing is most effective with large model sets and accurate cost and quality estimates. Furthermore, our new cascading strategy outperforms the threshold-based cascade by up to 2%, reinforcing the practical relevance of our theoretical results.

| | Three Models | | | Five Models | | | Eleven Models | | |
| --- | --- | --- | --- | --- | --- | --- | --- | --- | --- |
| | Low | Med | High | Low | Med | High | Low | Med | High |
| Linear Interp. | 69.62 | 69.62 | 69.62 | 69.22 | 69.22 | 69.22 | 70.51 | 70.51 | 70.51 |
| Routing | 79.73 | 74.97 | 71.81 | 81.24 | 74.43 | 71.33 | 83.25 | 74.63 | 72.67 |
| Cascade (Baseline) | 80.86 | 74.64 | 72.48 | 82.33 | 73.03 | 69.53 | 84.48 | 73.64 | 69.79 |
| Cascade (Ours) | 81.13 | 76.10 | 72.67 | 83.05 | 75.15 | 70.18 | 84.45 | 75.10 | 70.26 |
| Cascade Routing | **82.34** | **76.56** | **73.23** | **84.34** | **76.32** | **72.74** | **87.28** | **77.62** | **74.40** |

Table 1: AUC scores in % for different strategies on RouterBench across model and noise levels. Numbers for all baselines are always worse than the 95% confidence intervals of cascade routing. For a discussion on confidence intervals, we refer to App. E.

| | Classification | | | Open-Form | | |
| --- | --- | --- | --- | --- | --- | --- |
| | LLAMA | GEMMA | MISTRAL | LLAMA | GEMMA | MISTRAL |
| Linear Interp. | 74.28 | 61.68 | 63.39 | 79.11 | 54.10 | 53.86 |
| Routing | 74.92 | 64.44 | 64.89 | 79.32 | 58.40 | **58.71** |
| Cascade (Baseline) | 74.79 | 54.31 | 61.22 | 79.23 | 56.18 | 48.29 |
| Cascade (Ours) | 75.42 | 62.82 | 63.36 | 79.68 | 57.67 | 55.51 |
| Cascade Routing | **75.52** | **64.70** | **64.97** | **79.84** | **59.62** | 58.69 |

Table 2: AUC scores on a classification and open-form reasoning benchmark. Highest numbers are bolded, underlined numbers are within the 95% confidence intervals of the highest number. For a discussion on confidence intervals, refer to App. E. In App. H, we present benchmark-specific AUC values for each benchmark separately.

## 5.2 OTHER BENCHMARKS

RouterBench does not provide log probabilities associated with model answers, which constrains the construction of more realistic quality estimates using features like perplexity. To address this, we develop two benchmarks that better simulate practical use cases for cascade routing.

**Datasets** We perform experiments on classification and open-form reasoning tasks. The classification benchmarks include ARC-Challenge (Clark et al., 2018), MMLU-Pro (Wang et al., 2024), and MixEval (Ni et al., 2024). For open-form reasoning tasks, we use MMLU-Pro and GSM8k (Cobbe et al., 2021). In classification, models select a single option representing their answer, with no intermediate reasoning process. In contrast, open-form reasoning allows models to generate their answers after reasoning. We split each benchmark into a training set for training quality and cost estimates and a test set for evaluation. We evaluate nine models, three each from the LLAMA-3.1 (AI@Meta, 2024), GEMMA (Gemma Team et al., 2024), and MISTRAL (Jiang et al., 2023) model families. The exact models are specified in App. D.

**Quality and Cost Estimates** We estimate model quality by fitting a linear model based on features that reflect model uncertainty, such as perplexity (Gupta et al., 2024). Further features include the originating benchmark of each sample and whether earlier models in the cascade agree on their prediction. Full details on the features are provided in App. D. The linear model is trained using these features on the benchmark's training split.

For cost estimation, we first calculate the number of tokens in both the query and the model's response. We then use API-based prices per token for each model to estimate the cost.[3] In classification, where responses consist of a single token, the cost can be determined before running the model. In open-form reasoning tasks, where response lengths vary, we estimate this length based on responses from previous models in the cascade if the model has not yet been computed. If no model response is available, we estimate the response length using the average from the training data.

**Results** Table 2 presents the results for the LLAMA, GEMMA, and MISTRAL model families across both benchmarks. Cascade routing consistently performs on par with or outperforms all baselines,

---

[3]We used the Together API for all our experiments.

| | Low-Noise | | Medium-Noise | | High-Noise | |
|---|---|---|---|---|---|---|
| | AUC (%) | Time (ms) | AUC (%) | Time (ms) | AUC (%) | Time (ms) |
| Cascade Routing | 87.26 | 12.05 | 77.60 | 12.75 | 74.41 | 9.41 |
| SLOW | 87.29 | 80.04 | 77.62 | 87.23 | 74.41 | 67.45 |
| GREEDY | 85.93 | 1.64 | 77.16 | 1.59 | 74.35 | 1.04 |
| NO-EXPECT | 85.98 | 3.37 | 77.11 | 3.04 | 74.35 | 1.77 |

Table 3: AUC scores and average runtime for various variations of cascade routing on RouterBench when using all eleven models.

though with narrower margins reaching up to $1.2\%$. This reduced gain can be attributed to two main factors. First, the quality and cost estimates are very noisy, leading to performance gains over the naive baseline similar to those observed in high-noise scenarios on RouterBench. Second, the cascading strategy sometimes underperforms compared to the linear interpolation baseline, indicating that the post-computation features used for quality estimation, such as perplexity, do not provide sufficient advantage to warrant running models in a cascading fashion.

As expected, cascade routing is most effective when both routing and cascading outperform the linear interpolation baseline. When cascading offers no or very small performance improvement, cascade routing typically reduces to pure routing. Larger performance gains are observed only when cascading adds value beyond routing. Our results suggest that more sophisticated methods are needed to enhance quality estimates and improve the overall effectiveness of all model selection strategies in practical applications. In App. G, we analyze the latency of cascade routing compared to routing is acceptable for practical use cases.

### 5.3 ABLATION STUDY

We conduct an ablation study to examine the impact of various design choices in cascade routing on performance and runtime. Runtime is a critical factor because the overhead introduced by the strategy must be negligible compared to the time required for model computation. If the strategy adds significant overhead, its performance gains may be offset by the increased runtime. We also include an additional ablation that specifically targets runtime on random data in App. F.

To investigate this, we repeat the experiment from §5.1 when using all eleven models, testing different variations of cascade routing. We evaluate a slower variation that omits Lemma 1, thereby requiring more supermodels to be evaluated (SLOW), a greedy variation that only considers supermodels of length $j + 1$ at step $j$ (GREEDY), and a version that does not compute the expected value when evaluating supermodel quality, using the quality of the best model instead (NO-EXPECT).

**Results**   Table 3 presents the results of the ablation study. As expected, the SLOW variation is almost an order of magnitude slower while achieving similar performance. In contrast, both GREEDY and NO-EXPECT are faster but perform worse in the low- and medium-noise scenarios by $0.5\%$ to $1.3\%$. Interestingly, there seems to be a much smaller performance gap in the high-noise scenario. This is due to the very low variance in the quality estimates, since the linear model used for quality estimation predicts an almost constant value for each query in this scenario, making the expected value computation less important.

Furthermore, the GREEDY and NO-EXPECT variants perform very similarly, while GREEDY is about twice as fast as NO-EXPECT. This suggests that one should almost always use the normal variant of cascade routing, and only consider the GREEDY variant if runtime is a critical concern. Neither the SLOW nor the NO-EXPECT variant is recommended, as they either perform worse or are significantly slower than the normal variant.

## 6   RELATED WORK

We discuss related work in routing and cascading.

**Routing**   Routing is a widely studied problem in machine learning, particularly in the task of directing input queries to specialized expert models. One of the most common applications of routing is model selection for natural language input queries with a known correct answer (Ding et al., 2024; Hari and Thomson, 2023; Liu et al., 2024; Jang et al., 2023; Nguyen et al., 2024; Sakota et al., 2024; Shnitzer et al., 2023). All these works train a machine learning model to predict whether a given model will correctly answer a query. Though the setups in these works are largely similar, they vary in certain specifics, such as the type of input queries or the features used for quality estimation.

Routing is also applied in other areas. For instance, Lu et al. (2024); Ong et al. (2024) use preference data to train a quality estimator, which facilitates routing in scenarios involving real-world user queries where clear ground-truth answers may not exist. Additionally, Chen et al. (2022) employ routing for API selection in multi-label classification tasks, focusing on directing queries to the appropriate API based on task requirements. Similarly, Zhang et al. (2024) apply routing in software agent environments, directing user issues to the agent most suited to handle them.

**Cascading**   Cascading techniques are primarily used to reduce inference costs by employing smaller models initially and only cascading to larger models if the smaller ones fail to provide a sufficiently accurate answer. Most often, cascading decisions are based on the smaller model's confidence in its own predictions (Chen et al., 2023; Ramírez et al., 2024; Varshney and Baral, 2022). However, alternative techniques also exist. For example, Madaan et al. (2023) propose running models multiple times and measuring the variance in their responses to decide whether to cascade to a larger model.

For classification tasks, early stopping is another cascading strategy (Li et al., 2021; Schuster et al., 2022). In this approach, the cascade halts when a model's intermediate layers generate representations that are sufficiently informative to predict the correct class. This reduces computational costs by avoiding the need to process every query through the entire model.

There has also been specific research on quality estimation within cascading frameworks. Gupta et al. (2024) examine various measures of uncertainty in language model answers, evaluating their impact on cascading performance. Meanwhile, Jitkrittum et al. (2023) explore failure cases in cascading mechanisms that rely on uncertainty, introducing alternative quality measures that enhance cascade efficiency. Lastly, Xue et al. (2023) apply cascading to majority voting for a single model to obtain a method called dynamic voting: the cascade stops depending on the aggregated answers of all previous model computations. This allows the system to process simpler queries using fewer votes while allocating more computational resources for harder queries.

## 7   LIMITATIONS

While cascade routing provides a promising approach to improve model selection strategies, it requires accurate cost and quality estimates to be effective. However, in §5.2 we found that quality estimates based on current state-of-the-art methods contain significant noise. This limits the performance improvement of cascade routing over routing, as the quality estimates are not accurate enough to guide the selection of the best model. As shown in §5.1, lower noise in the quality and cost estimates lead to much better performance of all model selection strategies, indicating that further work that aims to reduce the noise in the quality and cost estimates is needed to fully leverage the potential of cascade routing.

Another limitation is the necessity of a training dataset to optimize the hyperparameters associated with cascade routing. While both routing and cascading suffer from the same disadvantage, the training data required for routing is much more minimal, since it only needs to estimate two hyperparameters. In contrast, cascade routing requires estimating multiple hyperparameters on a more complex optimization surface.

## 8   CONCLUSION

In this work, we introduced a novel framework for routing and cascading that enabled us to propose theoretically optimal strategies for both paradigms. Furthermore, we used this theoretical analysis to propose a new paradigm for model selection, cascade routing, which combines the benefits of both

routing and cascading. We showed that cascade routing can significantly outperform its baselines, especially with good quality and cost estimates. Furthermore, we found that our new cascading strategy significantly outperforms existing approaches to cascading, showing our theoretical analysis also leads to practical gains. Our work provides a theoretical foundation for model selection strategies and opens up new avenues for future research in this area, especially in the direction of obtaining more accurate quality and cost estimates.

## REPRODUCIBILITY STATEMENT

All our proofs are included in the appendix and contain the exact conditions under which each statement in the main text holds. We furthermore make our code available with clear instructions on how to reproduce our results.

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

# A  ROUTING

To prove Theorem 1, we first rewrite the routing optimization problem in Eq. (1) as a linear program over functions $s : \mathcal{X} \to \mathbb{R}^k$ instead of functions $s : \mathcal{X} \to \Delta_k$. This makes the optimization problem more tractable. Specifically, Eq. (1) can be rewritten as follows:

$$\max_r \quad \mathbb{E}_{x \sim \mathcal{X}} \left[ \sum_{i=1}^{k} s_i(x) \hat{q}_i(x) \right]$$

$$\text{s.t.} \quad \mathbb{E}_{x \sim \mathcal{X}} \left[ \sum_{i=1}^{k} s_i(x) \hat{c}_i(x) \right] \leqslant B \qquad (3)$$

$$\forall i \in \{1, ..., k\} : \forall x \in \mathcal{X} : s_i(x) \geq 0 \wedge \sum_{j=1}^{k} s_j(x) = 1$$

We then rewrite Theorem 1 to allow for a more exact formulation of the optimal routing strategy:

**Theorem 4.** *(Optimal Routing Strategy) Suppose there exists an admissible solution to the set of constraints in Eq. (3). For any $\lambda \in \mathbb{R}^+$, let $S_\lambda$ be the set of routing strategies $s$ that satisfy the following constraints:*

$$\forall x \in \mathcal{X}, \forall i \in \{1, ..., k\} : \hat{q}_i(x) - \lambda \hat{c}_i(x) < \max_j \hat{q}_j(x) - \lambda \hat{c}_j(x) \Rightarrow s_i(x) = 0 \qquad (4)$$

*If there exists a strategy in $S_0$ that has a cost less than or equal to $B$, then this strategy achieves the optimal quality. Otherwise, there exists a $\lambda^* \in \mathbb{R}^+$ such that $S_\lambda$ contains a routing strategy that has exactly cost $B$ and all routing strategies in $\bigcup_{\lambda \in \mathbb{R}^+} S_\lambda$ that have cost $B$ achieve the same optimal quality.*

There is one extra condition mentioned here that we omitted in the main text. The requirement of having at least an admissible solution to the constraints in Eq. (3) is necessary to ensure that the set of possible solutions to Eq. (3) is not empty. For instance, the cost budget $B$ can be too low such that even running the cheapest model for each query is too expensive.

The formulation of $s_{\text{OPT}}$ as a convex combination of $s_{\text{MIN}}^{\lambda}$ and $s_{\text{MAX}}^{\lambda}$ is a direct consequence of Theorem 4. Indeed, let $\lambda^*$ be as defined in Theorem 4. Then $s_{\text{MIN}}^{\lambda^*}$, resp. $s_{\text{MAX}}^{\lambda^*}$, must have the lowest, resp. highest, cost among all routing strategies in $S_{\lambda^*}$. Since there is a routing strategy in $S_{\lambda^*}$ that has cost $B$, there must exist a convex combination of $s_{\text{MIN}}^{\lambda^*}$ and $s_{\text{MAX}}^{\lambda^*}$ that also has cost $B$ and thus achieves the optimal quality.

We first prove several lemmas before proving the theorem.

**Lemma 2.** *$S_\lambda$ is non-empty and convex for all $\lambda \in \mathbb{R}^+$.*

*Proof.* Non-emptiness follows from the fact that the routing strategy that assigns all probability mass for a sample $x$ to a model $i$ for which $\hat{q}_i(x) - \lambda \hat{c}_i(x)$ is maximal, is in $S_\lambda$. For convexity, let $s^{(1)}, s^{(2)} \in S_\lambda$ be arbitrary. Let $s^\gamma$ be the convex combination of $s^{(1)}$ and $s^{(2)}$ with weight $\gamma \in [0, 1]$. Let $x \in \mathcal{X}$ be arbitrary. Then, $s_i^\gamma(x) > 0$ if and only if $s_i^{(1)}(x) > 0$ or $s_i^{(2)}(x) > 0$. Since $s^{(1)}, s^{(2)} \in S_\lambda$, we have $\hat{q}_i(x) - \lambda \hat{c}_i(x) \geqslant \max_j \hat{q}_j(x) - \lambda \hat{c}_j(x)$ for all $i$ such that $s_i^{(1)}(x) > 0$ or $s_i^{(2)}(x) > 0$. This implies that $\hat{q}_i(x) - \lambda \hat{c}_i(x) \geqslant \max_j \hat{q}_j(x) - \lambda \hat{c}_j(x)$ for all $i$ such that $s_i^\gamma(x) > 0$. Thus, $s^\gamma \in S_\lambda$. $\qquad \square$

**Lemma 3.** *Let $\lambda_1 < \lambda_2$ and $s^{(1)}$, resp. $s^{(2)}$ be arbitrary routing strategies in $S_{\lambda_1}$, resp. $S_{\lambda_2}$. Then, the cost of $s^{(1)}$ is greater or equal to the cost of $s^{(2)}$, i.e.,*

$$\mathbb{E}_{x \sim \mathcal{X}} \left[ \sum_{i=1}^{k} s_i^{(1)}(x) \hat{c}_i(x) \right] \geqslant \mathbb{E}_{x \sim \mathcal{X}} \left[ \sum_{i=1}^{k} s_i^{(2)}(x) \hat{c}_i(x) \right]$$

*Proof.* We show that for any $x \in \mathcal{X}$, the cost of $s^{(1)}$ is greater or equal to the cost of $s^{(2)}$. Let $x \in \mathcal{X}$ be arbitrary. Suppose $s^{(1)}$ is strictly cheaper than $s^{(2)}$. Then, there must exist a model pair $i, j$ such

that $\hat{c}_i(x) < \hat{c}_j(x)$, $s_i^{(1)}(x) > s_i^{(2)}(x) \geqslant 0$, and $s_j^{(2)}(x) > s_j^{(1)}(x) \geqslant 0$. However, $s_i^{(1)}(x) > 0$ implies

$$\hat{q}_i(x) - \lambda_1 \hat{c}_i(x) \geqslant \hat{q}_j(x) - \lambda_1 \hat{c}_j(x).$$

Furthermore, since $\lambda_1 - \lambda_2 < 0$, we have

$$\hat{c}_i(x)(\lambda_1 - \lambda_2) > \hat{c}_j(x)(\lambda_1 - \lambda_2).$$

Adding these two inequalities gives

$$\hat{q}_i(x) - \lambda_2 \hat{c}_i(x) > \hat{q}_j(x) - \lambda_2 \hat{c}_j(x),$$

which is a contradiction with $s_j^{(2)}(x) > 0$. Thus, the cost of $s^{(1)}$ is greater or equal to the cost of $s^{(2)}$. □

**Lemma 4.** *Let $\Lambda$ be the set of points $\lambda \in \mathbb{R}$ such that there exist an $x \in \mathcal{X}$ and $i \neq j$ such that $\hat{q}_i(x) - \lambda \hat{c}_i(x) = \hat{q}_j(x) - \lambda \hat{c}_j(x)$. Let $\lambda_1 < \lambda_2$ be such that $[\lambda_1, \lambda_2] \cap \Lambda = \emptyset$. Then, $S_{\lambda_1} = S_{\lambda_2}$. Furthermore, if $[\lambda_1, \lambda_2] \cap \Lambda = \{\lambda^*\}$, then $S_\lambda \subset S_{\lambda^*}$ for all $\lambda \in [\lambda_1, \lambda_2]$.*

*Proof.* We first show the first statement by showing that $S_{\lambda_1} \setminus S_{\lambda_2} = \emptyset$. $S_{\lambda_2} \setminus S_{\lambda_1} = \emptyset$ follows analogously. Suppose there exists a routing strategy $s \in S_{\lambda_1} \setminus S_{\lambda_2}$. Since $s \notin S_{\lambda_2}$, there must exist an $x \in \mathcal{X}$ and model $i$ such that $s_i(x) > 0$ and $\hat{q}_i(x) - \lambda_2 \hat{c}_i(x) < \max_j \hat{q}_j(x) - \lambda_2 \hat{c}_j(x)$. Let $j$ be an index such that $\hat{q}_i(x) - \lambda_2 \hat{c}_i(x) < \hat{q}_j(x) - \lambda_2 \hat{c}_j(x)$. Since $s \in S_{\lambda_1}$, we have $\hat{q}_i(x) - \lambda_1 \hat{c}_i(x) \geqslant \hat{q}_j(x) - \lambda_1 \hat{c}_j(x)$. By continuity, there exists a $\lambda \in [\lambda_1, \lambda_2]$ such that $\hat{q}_i(x) - \lambda \hat{c}_i(x) = \hat{q}_j(x) - \lambda \hat{c}_j(x)$, which is a contradiction with $[\lambda_1, \lambda_2] \cap \Lambda = \emptyset$.

Now suppose $[\lambda_1, \lambda_2] \cap \Lambda = \{\lambda^*\}$. Let $\lambda \in [\lambda_1, \lambda^*)$ be arbitrary and let $s \in S_\lambda$ be arbitrary. We show that $s \in S_{\lambda^*}$. For $\lambda \in (\lambda^*, \lambda_2]$, the proof is completely analogous. By contradiction, suppose there exists an $x \in \mathcal{X}$ and model $i$ such that $s_i(x) > 0$ and $\hat{q}_i(x) - \lambda^* \hat{c}_i(x) < \max_j \hat{q}_j(x) - \lambda^* \hat{c}_j(x)$. This means there exists a model $j$ such that $\hat{q}_i(x) - \lambda^* \hat{c}_i(x) < \hat{q}_j(x) - \lambda^* \hat{c}_j(x)$. Since $s \in S_\lambda$, we know that $\hat{q}_i(x) - \lambda \hat{c}_i(x) \geqslant \hat{q}_j(x) - \lambda \hat{c}_j(x)$. This implies that there must exist a $\lambda' \in [\lambda_1, \lambda^*)$ such that $\hat{q}_i(x) - \lambda' \hat{c}_i(x) = \hat{q}_j(x) - \lambda' \hat{c}_j(x)$. However, this is a contradiction with $[\lambda_1, \lambda^*) \cap \Lambda = \emptyset$. Thus, $s \in S_{\lambda^*}$. □

In what follows, we will assume that $|\Lambda| < \infty$. This is a very minor assumption. For instance, if $\hat{q}$ and $\hat{c}$ only take on a finite amount of values, this is trivially satisfied. Since estimators are implemented on a computer, they will always have a finite precision, meaning that $\hat{q}$ and $\hat{c}$ will only take on a finite amount of values.

**Lemma 5.** *Let $\lambda_1 < \lambda_2$ and $s^{(1)}$, resp. $s^{(2)}$ be arbitrary routing strategies in $S_{\lambda_1}$, resp. $S_{\lambda_2}$, with costs resp. $B_1$ and $B_2$. Then, for any $B \in [B_1, B_2]$ there exists a $\lambda \in [\lambda_1, \lambda_2]$ such that $S_\lambda$ contains a routing strategy that has exactly cost $B$.*

*Proof.* Let $B \in [B_1, B_2]$ be arbitrary. If $B = B_1$ or $B = B_2$, the statement is trivially true. Therefore, suppose $B \in (B_1, B_2)$. Let $\Lambda$ be as defined in Lemma 4. By Lemma 3, there exists a $\lambda^* \in [\lambda_1, \lambda_2]$ such that all strategies in $S_\lambda$ for $\lambda < \lambda^*$, resp. $\lambda > \lambda^*$, have cost at least, resp. at most, $B$. If $\lambda^* \notin \Lambda$, then the first part of Lemma 4, together with $|\Lambda| < \infty$, implies that $S_{\lambda^*} = S_{\lambda^* - \epsilon} = S_{\lambda^* + \epsilon}$ for some $\epsilon > 0$. All the strategies in $S_{\lambda^*}$ must therefore have cost both at least and at most $B$, meaning they should equal $B$. We can therefore assume that $\lambda^* \in \Lambda$. By Lemma 4 and $|\Lambda| < \infty$, there is en $\epsilon > 0$ such that $S_{\lambda^* - \epsilon} \subset S_{\lambda^*}$ and $S_{\lambda^* + \epsilon} \subset S_{\lambda^*}$. Let $s^- \in S_{\lambda^* - \epsilon}$ and $s^+ \in S_{\lambda^* + \epsilon}$ be arbitrary. Let $s^\gamma$ be the convex combination of $s^-$ and $s^+$ with weight $\gamma \in [0, 1]$. Since $s^-, s^+ \in S_{\lambda^*}$, we have $s^\gamma \in S_{\lambda^*}$ by Lemma 2. Denote by $B^-$, resp. $B^+$, the cost of $s^-$, resp $s^+$. Furthermore, the cost of $s^\gamma$ is $\gamma B^- + (1 - \gamma) B^+$. Since $B \in [B^-, B^+]$, there exists a $\gamma \in [0, 1]$ such that $s^\gamma$ has cost exactly $B$. □

We can now prove the theorem.

*Proof.* If $S_0$ contains a solution that has cost less than or equal to $B$, then this solution trivially achieves the optimal quality. Thus, for the rest of the proof we can assume that the cost of every solution in $S_0$ is greater than $B$. For $\lambda \to \infty$, $S_\lambda$ contains the solution that assigns all probability mass

to the model with the lowest cost. Since there is an admissible solution, this solution necessarily has cost less than $B$. Therefore, by Lemma 5, there exists a $\lambda^* \in \mathbb{R}$ such that $S_{\lambda^*}$ contains a routing strategy that has exactly cost $B$.

Let $s$ be an arbitrary routing strategy in $\bigcup_{\lambda \in \mathbb{R}^+} S_\lambda$ that has cost $B$. Specifically, let $s \in S_\lambda$. Let $s'$ be any other routing strategy that is an admissible solution to the optimization problem. Then:

$$
\begin{aligned}
\mathbb{E}_{x \in X}\left[\sum_{i=1}^{k} s_i'(x)\hat{q}_i(x)\right] &= \mathbb{E}_{x \in X}\left[\sum_{i=1}^{k} s_i'(x)\hat{q}_i(x) - \lambda B + \lambda B\right] \\
&\leqslant \mathbb{E}_{x \in X}\left[\sum_{i=1}^{k} s_i'(x)\left(\hat{q}_i(x) - \lambda\hat{c}_i(x)\right) + \lambda B\right] \\
&\leqslant \mathbb{E}_{x \in X}\left[\sum_{i=1}^{k} s_i(x)\left(\hat{q}_i(x) - \lambda\hat{c}_i(x)\right) + \lambda B\right] \\
&= \mathbb{E}_{x \in X}\left[\sum_{i=1}^{k} s_i(x)\hat{q}_i(x)\right]
\end{aligned}
$$

Thus, $s$ achieves the optimal quality.

$\square$

## B  CASCADING

To prove Theorem 2, we heavily rely on the results derived in App. A. As explained in §3, cascading can be reinterpreted as a sequence of routing problems. However, to prove optimality, we need to be slightly more careful with the exact formulation of the problem.

At step $j$, the cascading strategy needs to decide whether to stop the cascade or to continue to the next model. It should continue to the next model if any of the supermodels $M_{1:j}, \ldots, M_{1:k}$ is better to run than $M_{1:j-1}$ for some measure of 'better'. Therefore, the cascading strategy is indeed performing a routing operation between the supermodels $M_{1:j-1}, \ldots, M_{1:k}$.

However, the optimization problem does slightly change compared to the routing problem. First of all, for each query $x \in \mathcal{X}$, there is a possibility that the cascade is stopped before step $j$. Therefore, the cascade should not aim to optimize the quality at step $j$ for such a query, since it would not have any effect on the overall quality of the cascade. Furthermore, the budget $B$ is only enforced over the entire cascade, and not over the individual steps. Since the problem changes through steps, it is not required that the cost of the router at step $j$ is exactly equal to $B$.

Therefore, we reformulate cascading using an inner and outer optimization problem. The inner optimization problem aims to find the optimal routing strategy at step $j$ for a given budget $B_j$. The outer optimization problem aims to find the optimal budget $B_j$ for each step $j$ such that the overall quality of the cascade is maximized under the constraint that the total cost of the cascade is at most $B$.

To formulate this more exactly, let $P_j(M)$ be the probability that the cascade computed supermodel $M$ by step $j$. Then, the inner optimization problem at step $j$ can be formulated as:

$$\max_{r^{(j)}} \quad \mathbb{E}_{x \sim \mathcal{X}} \left[ P_j(M_{1:j-1}) \sum_{i=j-1}^{k} r_{1:i}(x) \hat{q}_{1:i}^{(j)}(x) \right]$$

$$\text{s.t.} \quad \mathbb{E}_{x \sim \mathcal{X}} \left[ P_j(M_{1:j-1}) \sum_{i=j-1}^{k} r_{1:i}(x) \hat{c}_{1:i}^{(j)}(x) \right] \leqslant B_j \tag{5}$$

$$\forall i \in \{j-1, ..., k\} : \forall x \in \mathcal{X} : r_{1:i}(x) \geq 0 \wedge \sum_{i=j-1}^{k} r_{1:i}(x) = 1$$

Note that $P_j(M_{1:j-1})$ can be incorporated in the quality and cost estimates. This leaves us with the exact same optimization problem as the routing problem, but with a different budget $B_j$. Since the chosen model only depends on the maximization of $P_j(M_{1:j-1})\hat{q}_i^{(j)}(x) - \lambda_j P_j(M_{1:j-1})\hat{c}_i^{(j)}(x)$, the probability $P_j(M_{1:j-1})$ can be divided out of the optimization problem.

The inner optimization problems prove the existence of optimal routing strategies at each step $j$ with parameters $\lambda_j$. We note that there only needs to be one parameter $\gamma$ that determines the convex combination since the budget $B$ is only enforced over the entire cascade.

Let us denote the quality and cost of the entire cascading strategy for given parameters $\lambda_1, \ldots, \lambda_k$ and $\gamma$ as $Q(\lambda_1, \ldots, \lambda_k, \gamma)$ and $C(\lambda_1, \ldots, \lambda_k, \gamma)$ respectively. Then, the outer optimization problem can be formulated as:

$$\max_{\lambda_1, \ldots, \lambda_k, \gamma} \quad Q(\lambda_1, \ldots, \lambda_k, \gamma)$$
$$\text{s.t.} \quad C(\lambda_1, \ldots, \lambda_k, \gamma) \leqslant B \tag{6}$$

To solve this outer optimization problem, we simply perform a hyperparameter search over the budgets $B_1, \ldots, B_k$ using a hyperparameter optimization search as discussed in §3.

### B.1 PRIOR APPROXIMATIONS

We now prove Corollary 1. Before doing so, we first need to define what we exactly mean by equivalency. For this purpose, let $\mathcal{C}_1$ be defined as follows:

$$\mathcal{C}_1 = \left\{ s \mid s \text{ is a cascading strategy with parameters } \lambda_1, \ldots, \lambda_k, \gamma = 0 \text{ using estimates } \hat{q}^{(j)}, \hat{c}^{(j)} \right\}$$

Similarly, let $\mathcal{C}_2$ be defined as follows:

$$\mathcal{C}_2 = \left\{ s \mid s \text{ is a thresholding strategy with parameters } \tau_1, \ldots, \tau_k \text{ using estimates } \hat{q}^{(j)}, \hat{c}^{(j)} \right\}$$

We note that we set $\gamma = 0$ since the thresholding strategy is deterministic. We therefore restrict the cascading strategy to be deterministic as well.

We define the equivalence between the two sets as follows:

**Definition 6** (Equivalence of Strategies). *We say a set of strategies $\mathcal{C}_1$ is equivalent to another set of strategies $\mathcal{C}_2$, denoted as $\mathcal{C}_1 \equiv \mathcal{C}_2$, if for all $s_0 \in \mathcal{C}_1 \cup \mathcal{C}_2$ there exists a $s_1 \in \mathcal{C}_1$, and a $s_2 \in \mathcal{C}_2$ such that for all $x \in \mathcal{X}$, $s_0$, $s_1$ and $s_2$ take the same decisions on $x$.*

We can now more accurately state the conditions under which the thresholding strategy is equivalent to the optimal strategy.

**Corollary 2** (Optimal Thresholding Strategy). *Let $\mathcal{C}_1, \mathcal{C}_2$ be defined as above. Then, $\mathcal{C}_1 \equiv \mathcal{C}_2$ if and only if there exists alternative quality and cost estimates $\hat{q}_i^{(j)'}(x)$ and $\hat{c}_i^{(j)'}(x)$ with associated set of cascading strategies $\mathcal{C}_1'$ such that $\mathcal{C}_1 \equiv \mathcal{C}_1'$ and the following conditions hold on these alternative quality and cost estimates: $\hat{c}_i^{(j)'}(x)$ is independent of $x$ and bigger than $0$, $\hat{q}_i^{(j)'}(x)$ is independent of $x$ for all $i \geqslant j$, and $\hat{q}_{1:i}^{(j)'}(x)$ is equal to $\hat{q}_i^{(j)'}(x)$.*

The main difference between Corollary 2 and Corollary 1 is that we impose the possibility of alternative quality and cost estimates. However, this does not really influence equivalency in the intuitive sense. Indeed, one could alternatively phrase the corollary as follows: the thresholding strategy is equivalent to any of our cascading strategies if and only if it is possible to construct alternative estimates such that the conditions hold.

*Proof.* We note that the cascade $s \in \mathcal{C}_1$ continues on a sample if the following condition holds:

$$\hat{q}^{(j)}_{1:j-1}(x) - \lambda_j \hat{c}^{(j)}_{1:j-1}(x) < \max_{i \in \{j,...,k\}} \hat{q}^{(j)}_{1:i}(x) - \lambda_j \hat{c}^{(j)}_{1:i}(x) \tag{7}$$

If $\mathcal{C}_1 \equiv \mathcal{C}'_1$, it is clear that Eq. (7) reduces to the thresholding strategy for all strategies in $\mathcal{C}'_1$. Indeed, for any $s \in \mathcal{C}'_1$, set $\tau_j = \max_{i \in \{j,...,k\}} \hat{q}^{(j)}_{1:i} - \lambda_j \hat{c}^{(j)}_{j:i}$ and the thresholding strategy is equivalent to $s$. Alternatively, if $s \in \mathcal{C}_2$, suppose $\max_{i \in \{j,...,k\}} \hat{q}^{(j)}_{1:i} - \lambda_j \hat{c}^{(j)}_{j:i} = \hat{q}^{(j)}_{1:i} - \lambda_j \hat{c}^{(j)}_{j:i}$ for some index $i$. Then, set $\lambda_j = \tau_j / \hat{c}^{(j)}_{j:i} - \hat{q}^{(j)}_{1:i} / \hat{c}^{(j)}_{j:i}$ and the cascading strategy is equivalent to $s$. Therefore, $\mathcal{C}_1 \equiv \mathcal{C}'_1 \equiv \mathcal{C}_2$.

Suppose now that $\mathcal{C}_1 \equiv \mathcal{C}_2$. We construct alternative quality and cost estimates $\hat{q}^{(j)'}_i(x)$ and $\hat{c}^{(j)'}_i(x)$ such that the conditions hold and such that $\mathcal{C}_1 \equiv \mathcal{C}'_1$. For this purpose, we define $\hat{c}^{(j)'}_i(x) = 1$ for all $i, j \in \{1, \ldots, k\}$, $\hat{q}^{(j)'}_i(x) = 1$ for all $i \geqslant j$, and $\hat{q}^{(j)'}_i(x) = \hat{q}^{(j)}_i(x)$ otherwise. Furthermore, we set $\hat{q}^{(j)'}_{1:i}(x) = \hat{q}^{(j)'}_i(x)$ for all $i, j \in \{1, \ldots, k\}$. The equivalence of $\mathcal{C}'_1$ and $\mathcal{C}_2$ can now be proven analogously to the previous paragraph. Therefore, $\mathcal{C}_1 \equiv \mathcal{C}'_1 \equiv \mathcal{C}_2$. $\square$

## C CASCADE ROUTING

We first note that the proof of the optimality of the cascade routing strategy is equivalent to the proof of the optimality of the cascade strategy, except that the expectation in the optimization problem Eq. (5) is now not only over $x \in X$, but also over all possible supermodels that were computed by step $j - 1$. However, this does not change the optimization problem, and the proof is completely analogous to the proof given in §3. Thus, all we need to prove is Lemma 1. To prove the lemma, we first prove the following lemma.

**Lemma 6.** *Let $Q_1, ..., Q_k$ be distributions. Let $\mathcal{S}$ be the superset of $\{1, ..., k\}$. Then $f : \mathcal{S} \to \mathbb{R}$ defined as $f(S) = \mathbb{E}(\max_{i \in S} Q_i)$ is submodular. Here, we define $\max_{i \in \emptyset} Q_i = -\infty$*

*Proof.* Let $T \subset S \subset \{1, \ldots, k\}$ and $j \in \{1, \ldots, k\}$ be arbitrary. To show the submodularity of $f$, we need to show that

$$f(T \cup \{j\}) - f(T) \geq f(S \cup \{j\}) - f(S).$$

We can write:

$$\begin{aligned}
f(S \cup \{j\}) - f(S) &= \mathbb{E}(\max_{i \in S \cup \{j\}} Q_i) - \mathbb{E}(\max_{i \in S} Q_i) \\
&= \mathbb{E}(\max(0, Q_j - \max_{i \in S} Q_i)) \\
&\leqslant \mathbb{E}(\max(0, Q_j - \max_{i \in T} Q_i)) \\
&= \mathbb{E}(\max_{i \in T \cup \{j\}} Q_i) - \mathbb{E}(\max_{i \in T} Q_i) \\
&= f(T \cup \{j\}) - f(T).
\end{aligned}$$

In the proof, we needed $\max_{i \in \emptyset} Q_i = -\infty$ in the case $T = \emptyset$. $\square$

We note that the assertion that $\max_{i \in \emptyset} Q_i = -\infty$ corresponds to the fact that giving no answer to a query has $-\infty$ quality.

We can now prove Lemma 1.

|  | Quality | | Cost | |
|---|---|---|---|---|
|  | $\sigma_{\text{before}}$ | $\sigma_{\text{after}}$ | $\sigma_{\text{before}}$ | $\sigma_{\text{after}}$ |
| LOW | 0.6 | 0.3 | 0.0002 | 0.00005 |
| MEDIUM | 1.6 | 0.8 | 0.0004 | 0.0001 |
| HIGH | 2.4 | 1.2 | 100 | 100 |

Table 4: Standard deviations of the noise levels on the RouterBench dataset.

*Proof.* Let $M$ and $m$ be as in the lemma. Suppose $M'$ is a supermodel that contains all models in $M$. Furthermore, let $M'' = M' \setminus m$. We show that the supermodel $M''$ is always strictly preferred over $M'$. To see this, we note that the difference between $\tau_{M'}(x, \lambda)$ and $\tau_{M''}(x, \lambda)$ is equal to

$$\mathbb{E}(\max_{m' \in M'} \hat{q}_{m'}(x)) - \mathbb{E}(\max_{m' \in M''} \hat{q}_{m'}(x)) - \lambda_j \hat{c}_m(x)$$

By Lemma 6, this difference is smaller than $\hat{q}_M(x) - \hat{q}_{M \setminus \{m\}}(x) - \lambda_j \hat{c}_m(x)$. Thus, by assumption, this difference is negative, and therefore $M''$ is always preferred over $M'$, which concludes the proof. $\square$

## D  EXPERIMENTAL DETAILS

We describe some additional details about the experimental setup and the datasets used in our experiments.

### D.1  ROUTERBENCH

**Data Split**  We use $5\%$ of the RouterBench data (around 2000 samples) to optimize the hyperparameters of cascading, routing, and cascade routing. The remaining $95\%$ is used for evaluation. We use the same data split for all noise levels.

**Noise**  In Table 4 we specify the standard deviations of the noise levels on the RouterBench dataset. To put these numbers into context, we note that quality varies between $0$ and $1$, and the average cost of the smallest models is $0.000073$, while the average cost of the largest models is $0.003281$. We fit a logistic regression model on this noisy signal to obtain the quality and cost estimates. This simulates the noise in the features that are used to estimate the quality and cost of the models.

**Models**  In the evaluated scenarios for three models, we use the models MIXTRAL-8X7B-CHAT, GPT-3.5-TURBO-1106, and GPT-4-1106-PREVIEW. When using five models, we add WIZARDLM-13B-V1.2 and CLAUDE-V2 to the mix. For eleven models, we use all models available in the benchmark.

### D.2  OTHER BENCHMARKS

**Data Split**  We split each dataset in each benchmark into a training set and a test set, each comprising $50\%$ of the data. For all datasets except GSM8k, the training set is created by splitting the original test data. In the case of GSM8k, since a separate training set is already available, we use this pre-existing training data, leaving the original test set unchanged. The training set is then further divided, with $50\%$ used for training quality and cost estimators, and the remaining $50\%$ reserved for hyperparameter optimization through validation.

**Evaluation Setting**  We use completion-based evaluation in a one-shot setting for each benchmark. For the classification tasks, we obtain the probability associated with each class ("A", "B", "C", . . . ) from the model directly. For open-form reasoning tasks, we extract the answer by instruction the model to generate a completion that ends with an extractable answer. If the model does not output an answer in the correct format, we perform a best-effort extraction by trying various regex patterns. Details on the prompts and regex patterns used for each benchmark are provided in the code repository.

| | Three Models | | | Five Models | | | Eleven Models | | |
|---|---|---|---|---|---|---|---|---|---|
| | Low | Med | High | Low | Med | High | Low | Med | High |
| Cascade Routing | $82.37_{\pm0.16}$ | $76.57_{\pm0.18}$ | $73.22_{\pm0.20}$ | $84.33_{\pm0.15}$ | $76.32_{\pm0.18}$ | $72.75_{\pm0.19}$ | $87.24_{\pm0.13}$ | $77.58_{\pm0.17}$ | $74.41_{\pm0.18}$ |
| — Routing | $2.64_{\pm0.14}$ | $1.59_{\pm0.14}$ | $1.40_{\pm0.16}$ | $3.09_{\pm0.15}$ | $1.88_{\pm0.16}$ | $1.41_{\pm0.17}$ | $4.00_{\pm0.19}$ | $2.93_{\pm0.20}$ | $1.73_{\pm0.19}$ |
| — Cascade (Baseline) | $1.50_{\pm0.12}$ | $1.91_{\pm0.19}$ | $0.74_{\pm0.19}$ | $2.00_{\pm0.15}$ | $3.29_{\pm0.26}$ | $3.22_{\pm0.25}$ | $2.76_{\pm0.14}$ | $3.93_{\pm0.27}$ | $4.62_{\pm0.27}$ |
| — Cascade (Ours) | $1.28_{\pm0.11}$ | $0.39_{\pm0.14}$ | $0.54_{\pm0.18}$ | $1.27_{\pm0.11}$ | $1.15_{\pm0.21}$ | $2.57_{\pm0.24}$ | $2.77_{\pm0.13}$ | $2.47_{\pm0.22}$ | $4.15_{\pm0.26}$ |

Table 5: AUC scores in % for different strategies on RouterBench in the 0-shot setting across model and noise levels with $2\sigma$ confidence intervals.

| | Three Models | | | Five Models | | | Eleven Models | | |
|---|---|---|---|---|---|---|---|---|---|
| | Low | Med | High | Low | Med | High | Low | Med | High |
| Cascade Routing | $82.37_{\pm0.16}$ | $76.57_{\pm0.18}$ | $73.22_{\pm0.20}$ | $84.33_{\pm0.15}$ | $76.32_{\pm0.18}$ | $72.75_{\pm0.19}$ | $87.24_{\pm0.13}$ | $77.58_{\pm0.17}$ | $74.41_{\pm0.18}$ |
| — Routing | $2.31_{\pm0.13}$ | $1.64_{\pm0.16}$ | $1.10_{\pm0.14}$ | $3.08_{\pm0.16}$ | $1.94_{\pm0.16}$ | $1.21_{\pm0.14}$ | $3.44_{\pm0.17}$ | $3.13_{\pm0.21}$ | $1.60_{\pm0.17}$ |
| — Cascade (Baseline) | $0.64_{\pm0.10}$ | $0.28_{\pm0.14}$ | $0.22_{\pm0.17}$ | $1.23_{\pm0.12}$ | $2.19_{\pm0.20}$ | $2.83_{\pm0.23}$ | $1.64_{\pm0.13}$ | $2.29_{\pm0.24}$ | $3.09_{\pm0.26}$ |
| — Cascade (Ours) | $1.02_{\pm0.09}$ | $\mathbf{0.09_{\pm0.11}}$ | $\mathbf{0.10_{\pm0.15}}$ | $1.25_{\pm0.09}$ | $1.59_{\pm0.17}$ | $2.45_{\pm0.21}$ | $2.06_{\pm0.11}$ | $2.22_{\pm0.20}$ | $2.95_{\pm0.24}$ |

Table 6: AUC scores in % for different strategies on RouterBench in the 5-shot setting across model and noise levels with $2\sigma$ confidence intervals. Bold numbers indicate that the confidence interval contains zero.

**Models** For the LLAMA-3.1 model family, we use the models LLAMA-3.1-8B-INSTRUCT, LLAMA-3.1-70B-INSTRUCT, and LLAMA-3.1-405B-INSTRUCT. For the GEMMA model family, we use the models GEMMA-2B-INSTRUCT, GEMMA-2-9B-INSTRUCT, and GEMMA-2-27B-INSTRUCT. For the MISTRAL model family, we use the models MISTRAL-7B-INSTRUCT-V0.3, MIXTRAL-8X7B-INSTRUCT-V0.1, and MIXTRAL-8X22B-INSTRUCT-V0.1.

**Features Quality Estimates** We specify the exact features used for the logistic regression model that serves as the quality estimator in §5.2. First, we include a one-hot encoding of the various datasets in each benchmark. Furthermore, for classification, we include the probability associated with the highest class and the entropy of the class probabilities if the model has been computed. If several models have been computed, we include both whether they agree on their prediction, and the JS-divergence between their class probabilities. For open-form reasoning, we include the perplexity, number of tokens, and several quantiles of the logits if the model has been computed, in accordance with Gupta et al. (2024). If several models have been computed, we also include whether they agree on their prediction.

We note that we train a separate logistic regression model for each history of computed models, and for each model separately as well. Thus we have one linear model for each combination of a target model $m_i$ and computed models $m_{i_1}, \ldots, m_{i_j}$. All the linear models are trained on the training set included in the benchmark.

# E CONFIDENCE INTERVALS

To check whether the results obtained by cascade routing are significantly higher than our baselines in Tables 1, 2 and 11, we perform bootstrapping on the samples in the dataset. Specifically, we compute the confidence interval associated with the difference between the AUC scores of cascade routing and the baselines. If this difference is positive and its $2\sigma$ confidence interval does not contain zero, we can conclude that cascade routing is significantly better than the baseline. These confidence intervals are reported in Tables 5–7.

# F RUNTIME ANALYSIS

We further analyze the runtime of the four variants of cascade routing presented in §5.3. Specifically, we perform experiments with random data, scaling the number of models to 80 to evaluate the runtime of all variants. Furthermore, we include a fifth variant of cascade routing in the analysis MAX-DEPTH, which restricts cascade routing to a maximum depth of 3 models. MAX-DEPTH does not reduce performance of cascade routing if the optimal depth is less than or equal to 3 models. However, it does significantly reduce the runtime of cascade routing.

| | Classification | | | Open-Form | | |
|---|---|---|---|---|---|---|
| | LLAMA | GEMMA | MISTRAL | LLAMA | GEMMA | MISTRAL |
| Cascade Routing | $75.52_{\pm 0.64}$ | $64.85_{\pm 0.72}$ | $64.98_{\pm 0.70}$ | $79.92_{\pm 0.67}$ | $59.70_{\pm 0.82}$ | $58.78_{\pm 0.75}$ |
| − Routing | $0.59_{\pm 0.33}$ | $\mathbf{0.38}_{\pm \mathbf{0.49}}$ | $\mathbf{0.08}_{\pm \mathbf{0.11}}$ | $0.56_{\pm 0.40}$ | $1.26_{\pm 0.56}$ | $\mathbf{0.02}_{\pm \mathbf{0.05}}$ |
| − Cascade (Baseline) | $0.70_{\pm 0.30}$ | $10.51_{\pm 0.60}$ | $3.78_{\pm 0.77}$ | $0.65_{\pm 0.25}$ | $3.47_{\pm 0.39}$ | $10.45_{\pm 1.14}$ |
| − Cascade (Ours) | $\mathbf{0.06}_{\pm \mathbf{0.17}}$ | $2.04_{\pm 0.31}$ | $1.66_{\pm 0.43}$ | $0.20_{\pm 0.19}$ | $2.00_{\pm 0.27}$ | $3.06_{\pm 0.66}$ |

Table 7: AUC scores on a classification and open-form reasoning benchmark with $2\sigma$ confidence intervals. Bold numbers indicate that the confidence interval contains zero.

For each number of models, we generate 100 data points, each with random quality and cost estimates associated with each model. For each point, we generate the hyperparameters $\lambda_1, ..., \lambda_k$ and $\gamma$ randomly. We then report the average runtime of the five variants of cascade routing in Fig. 3.

The results show the varying computational complexity of the different variants of cascade routing. SLOW has the highest runtime, and becomes computationally too expensive even when using less than 20 models. In contrast, standard cascade routing has a significantly lower runtime, and is able to handle up to 40 models within a 1 second runtime. Its faster variant, MAX-DEPTH, is able to handle up to 80 models within a 1 second runtime. Furthermore, we now also see a clear difference between NO-EXPECT and GREEDY. While GREEDY remains computationally very cheap even for 80 models, NO-EXPECT has a significantly higher runtime, even obtaining higher runtimes than MAX-DEPTH for 80 models.

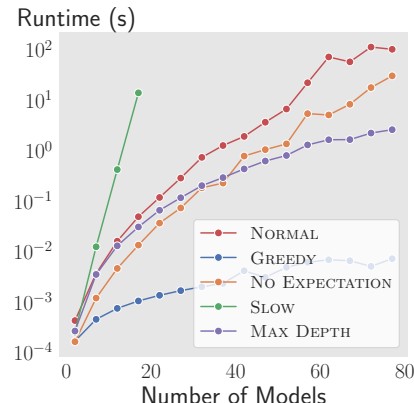

Figure 3: Runtime of the five variants of cascade routing for different numbers of models.

Thus, the conclusions from §5.3 are further supported by the runtime analysis: GREEDY is the most efficient variant of cascade routing, while NORMAL is the most efficient variant that does not compromise performance. MAX-DEPTH is a good choice if the optimal depth is known to be less than or equal to 3 models, as it significantly reduces runtime without compromising performance. Since cascades of more than 3 models are rare, MAX-DEPTH is a good choice in practice.

## G    LATENCY

We analyze the latency of cascade routing compared to routing to determine whether the additional computation time of cascade routing is justified by its improved quality-cost tradeoff. Since cascade routing involves running multiple models sequentially, we anticipate it will have a higher latency than routing.

In Table 8, we present the expected latencies for the classification and open-form reasoning benchmarks described in §5. Our analysis confirms that cascade routing does exhibit higher latency compared to routing, though the increase is less significant than expected. Specifically, across all benchmarks, the latency increase is capped at 0.6s. This relatively modest increase can be attributed to cascade routing's ability to terminate early when the supermodel achieves sufficiently high quality. Additionally, the lower latency of smaller models minimizes the overhead of running a smaller model before transitioning to a larger one.

## H    DETAILED RESULTS

We present benchmark-specific AUC values for the experiment performed in §5.2 in Table 9 for classification and Table 10 for open-form reasoning.

| | Classification | | | Open-Form | | |
|---|---|---|---|---|---|---|
| | LLAMA | GEMMA | MISTRAL | LLAMA | GEMMA | MISTRAL |
| Routing | 0.60 | 0.51 | 0.84 | 2.66 | 3.56 | 2.60 |
| Cascade (Baseline) | 0.91 | 0.74 | 0.67 | 3.65 | 4.11 | 3.92 |
| Cascade (Ours) | 1.07 | 0.89 | 0.94 | 4.17 | 4.41 | 4.24 |
| Cascade Routing | 0.81 | 0.64 | 0.85 | 3.24 | 4.06 | 2.74 |

Table 8: Averaged expected latencies on a classification and open-form reasoning benchmark.

| | LLAMA | | | GEMMA | | | MISTRAL | | |
|---|---|---|---|---|---|---|---|---|---|
| | MMLU | ARC | MixEval | MMLU | ARC | MixEval | MMLU | ARC | MixEval |
| Linear Interp. | 53.82 | 93.15 | 82.86 | 39.40 | 82.28 | 70.97 | 39.76 | 85.39 | 73.03 |
| Routing | 55.32 | 93.12 | 82.86 | 40.01 | 83.13 | 73.12 | 40.61 | 85.64 | 74.28 |
| Cascade (Baseline) | 54.80 | 94.08 | 84.15 | 36.43 | 77.53 | 66.10 | 36.99 | 83.88 | 72.73 |
| Cascade (Ours) | 55.05 | 94.16 | 84.00 | 37.68 | 79.80 | 70.57 | 37.03 | 86.27 | 74.42 |
| Cascade Routing | 55.40 | 93.90 | 83.91 | 39.93 | 83.74 | 73.16 | 40.56 | 86.52 | 74.64 |

Table 9: Classification AUC values for each benchmark separately for the experiment performed in §5.2

| | LLAMA | | GEMMA | | MISTRAL | |
|---|---|---|---|---|---|---|
| | MMLU | GSM8k | MMLU | GSM8k | MMLU | GSM8k |
| Linear Interp. | 65.64 | 94.43 | 36.52 | 73.86 | 41.40 | 67.84 |
| Routing | 65.75 | 94.15 | 38.08 | 75.01 | 43.03 | 68.00 |
| Cascade (Baseline) | 66.07 | 95.17 | 35.76 | 68.44 | 38.88 | 60.82 |
| Cascade (Ours) | 66.25 | 94.94 | 38.16 | 71.10 | 40.76 | 64.53 |
| Cascade Routing | 66.60 | 94.69 | 40.43 | 75.25 | 42.93 | 68.30 |

Table 10: Open form AUC values for each benchmark separately for the experiment performed in §5.2

| | Three Models | | | Five Models | | | Eleven Models | | |
|---|---|---|---|---|---|---|---|---|---|
| | Low | Med | High | Low | Med | High | Low | Med | High |
| Linear Interp. | 74.21 | 74.21 | 74.21 | 73.82 | 73.82 | 73.82 | 75.16 | 75.16 | 75.16 |
| Routing | 81.50 | 77.22 | 76.01 | 82.43 | 76.84 | 75.54 | 85.34 | 77.77 | 76.44 |
| Cascade (Baseline) | 83.16 | 78.58 | 76.89 | 84.27 | 76.59 | 73.92 | 87.14 | 78.60 | 74.94 |
| Cascade (Ours) | 82.68 | _78.79_ | _77.00_ | 84.26 | 77.20 | 74.30 | 86.67 | 78.67 | 75.08 |
| Cascade Routing | **83.82** | **78.92** | **77.11** | **85.51** | **78.82** | **76.74** | **88.78** | **80.88** | **78.02** |

Table 11: AUC scores in $\%$ for different strategies on RouterBench across model and noise levels for five-shot evaluation. Highest numbers are bolded, underlined numbers are within the $95\%$ confidence intervals of the highest number. For a discussion on confidence intervals, we refer to App. E.

## I    ADDITIONAL RESULTS

In Table 11 we report the AUC scores for the RouterBench dataset across different models and noise levels for the five-shot evaluation. Our conclusions presented in §5.1 remain consistent with the results presented in Table 11. However, there is one notable inconsistency: in two of the three low-noise scenarios, our cascading strategy performs worse than the threshold-based baseline cascade. In the scenario with three models, we find its cause can be found in the more difficult optimization surface for the hyperparameters of our cascading strategy. Specifically, our cascading strategy at some point starts to lose quality as cost increases. By simply setting the hyperparameters of the cascading strategy once it starts to lose quality to the ones where it obtained its highest quality, we obtain a quality of $83.35\%$ over the $83.17\%$ of the baseline cascade.

In contrast, for low-noise and eleven models, a similar approach does not yield a better result. Rather, the discrepancy is caused by a small mismatch between the quality estimates of supermodels and the chosen model. While the quality estimate is based on the expected maximum of all models, we restrict the selected model to be the last model that was computed in the cascade. Since the expected maximum is higher than the quality of the last model, this discrepancy can lead to suboptimal decisions. By allowing both the baseline cascade and our cascading strategy to select the model with the highest quality estimate, we find that our cascading strategy once again outperforms the baseline cascade. Note that this slight discrepancy is not relevant for cascade routing, since the extra restriction is not imposed in this setting.

