# OpenReview forum: "A Unified Approach to Routing and Cascading for LLMs"
_ICLR.cc/2025/Conference — Submitted to ICLR 2025_

### Official Review · Reviewer_tXch · 2024-10-25

**Soundness:** 2
**Presentation:** 2
**Contribution:** 2
**Rating:** 3
**Confidence:** 3

**Summary:**

The widespread applicability of large language models (LLMs) has increased the availability of many fine-tuned models of various sizes targeting specific tasks. An effective strategy can increase overall performance and even offer improvements over a single large monolithic model. Existing model selection strategies, such as routing and cascading, have their drawbacks. Routing commits to an initial model without flexibility, while cascading requires running every model in sequence, which can be inefficient. The paper proposes a novel approach called cascade routing, which combines the adaptability of routing with the cost-efficiency of cascading to improve output quality and reduce computational cost.

**Strengths:**

1. Good writing.
2. The research topic is valuable.
3. A formal proof is given.

**Weaknesses:**

1. The strategies proposed in the paper produced little improvement.
2. More solid experimentation is needed.

**Questions:**

1. The minor performance improvement (1%~4%) with the proposed scheme in the paper raises doubts about whether this enhancement is simply due to experimental uncertainty.  For greater conviction, I would advocate for more rigorous experimental designs that demonstrate substantial and statistically significant performance improvements.
2. What criteria should guide the decision on which models are assigned to the same cascade level? Additionally, what is the optimal number of models to be considered at each cascade level within the routing strategy？
3. Does combining cascading and routing hold the key to solving this problem, or would factors like quality estimation approaches be more promising topics for research?

---

> ### Author Response · Authors · 2024-11-21
>
> We thank the reviewer for their detailed review and questions. We are pleased to read that they find our paper well-written and the research topic valuable. Below, we address their questions and concerns.
>
> **Could you include statistical error bounds on your reported numbers?**
> Yes, we have now included a thorough analysis of statistical error bounds in Appendix F. Specifically, we underline values in Tables 1 and 2 that fall within the $2\sigma$ error bounds of the highest number. This analysis shows that, in Table 1, cascade routing consistently and significantly outperforms our baselines. In Table 2, cascade routing outperforms the baselines in three out of six cases and performs comparably in the other three cases.
>
> **Q2. Is the 1-4% accuracy improvement minor?**
> Please see Q2 of our main reply.
>
> **What criteria should guide the decision on which models are assigned to the same cascade level?**
> In our approach, all models are automatically assigned to every cascade level. Unlike traditional cascading methods, which require user-defined selection criteria, cascade routing dynamically determines the optimal selection strategy across models. This eliminates the need for restrictive pre-defined criteria, enabling a more flexible and efficient solution.
>
> **Does combining cascading and routing hold the key to solving this problem, or would factors like quality estimation approaches be more promising topics for research?**
> Both approaches are orthogonal and can significantly increase the performance of selection strategies. With our cascade routing approach we prove that, under our modeling assumptions, our given strategy is optimal. We therefore believe that research should now focus on more accurate quality estimators. Please refer to Q1 of our main reply for a full discussion on the impact of our optimal strategy and how it can help the community to shift focus to better quality estimators rather than algorithmic improvements.
>
> We hope that this addresses all the reviewers' questions and concerns. We are happy to discuss any further questions the reviewer might have.

---

> > ### Comment · Reviewer_tXch · 2024-11-26
> >
> > Thanks for your response. The performance improvement in Gupta et al. (2024) ranges from 1% to 19%, and the topics discussed in Jang et al. (2023) differ from those in this study. Additionally, the works of Ramirez et al. (2024) and Nguyen et al. (2024) have not yet been published. I still have concerns regarding the novelty and contribution of this work. Therefore, I maintain my rating.

---

> > > ### Author Response · Authors · 2024-11-27
> > >
> > > We thank the reviewer for their reply. We would like to clarify that the numbers reported by Gupta et al. in Table 1 represent **relative** improvements, whereas our results are expressed as **absolute** improvements. When considered in absolute terms, nearly all entries in their large table, except for five, show improvements below 4%, with a maximum improvement of 8% in a single entry. For instance, the 19% quoted by the reviewer is in fact just a 1.6% absolute improvement. Additionally, the majority of the larger improvements, including the 8%, are observed in the last column. It is important to note that the quality estimate in this last column requires running part of the large model, but the associated computational cost is not included in the reported improvements.

---

> > > > ### Author Response · Authors · 2024-11-29
> > > >
> > > > We thank the reviewer for taking the time to engage with our work. However, we are unclear about the basis of the concerns regarding the novelty and contribution of our work, which were not raised in the original review. To clarify the novel aspects and address any remaining doubts, we would like to emphasize the following key aspects of our paper that we believe represent significant advances in the field:
> > > > - **Deeper Understanding of Prior Approaches**: Our reformulation of both routing and cascading introduces a novel formulation that offers a more comprehensive understanding of their value. This reformulation allowed us to derive rigorous proofs of optimality and to propose a more effective approach to cascading. We believe this foundational advance provides both theoretical depth and practical utility, addressing gaps left by previous heuristic-based methods.
> > > > - **Development of Cascade Routing**: While cascade routing may appear as a straightforward generalization at first glance, its simplicity stems directly from our novel reformulation of cascading. Without this new formulation, it would not have been possible to formalize cascade routing in the rigorous manner presented in our work. In fact, the small proof required for cascade routing in Appendix C relies directly on the more complex proof presented in Appendix B.
> > > > - **Proof of Optimality and Its Implications**: As pointed out in Q1 in our main reply, by proving the optimality of our proposed algorithms, we establish that no significant gains can be made through further algorithmic improvements.
> > > >
> > > > We would appreciate it if the reviewer could clarify which specific aspects of these contributions are considered lacking in novelty or significance. If there are areas where additional explanation or evidence would be helpful, we would be happy to address them further.

---

### Official Review · Reviewer_pgV6 · 2024-11-02

**Soundness:** 4
**Presentation:** 4
**Contribution:** 3
**Rating:** 8
**Confidence:** 3

**Summary:**

This paper presents cascade routing, a novel model selection strategy for large language models (LLMs) that unifies the strengths of traditional routing and cascading methods. Routing directs queries to a specific model, optimizing for task-specific expertise but lacks flexibility to adjust once selected, while cascading processes queries sequentially through models, which can be inefficient. Cascade routing addresses these limitations by combining adaptability with cost-efficiency, dynamically routing queries across models until a high-quality response is reached. The authors theoretically derive optimal strategies for both routing and cascading, integrating them into the cascade routing framework, which is shown to significantly outperform existing methods in accuracy and cost-effectiveness across diverse benchmarks.

**Strengths:**

* Cascade routing effectively combines the strengths of routing and cascading for improved query handling in LLMs.

* The paper provides a solid theoretical basis, with optimal strategies derived for both routing and cascading.

* Experimental results confirm significant performance gains, especially under low-noise settings.

**Weaknesses:**

* Cascade routing's effectiveness is limited by the accuracy of quality and cost estimates, which can be noisy.

* Cascade routing requires multiple hyperparameters, making it challenging to tune effectively without extensive data.

**Questions:**

1. The cascade routing method heavily relies on precise estimates of quality and computational cost for each model. In practical scenarios, these estimates can be noisy, especially since they are typically derived from previous outputs or simple statistical models. When these estimates contain high levels of noise, cascade routing's decision-making becomes less effective. How can this issue be addressed especially for real-time or adaptive applications where quality and cost estimates are challenging to maintain with high accuracy?

2. Cascade routing involves tuning multiple hyperparameters to balance cost, quality, and routing decisions. This includes setting parameters for each model’s cost-quality tradeoff, which adds complexity to the optimization process. Would the optimization be time-consuming and computationally expensive?

---

> ### Author Response · Authors · 2024-11-21
>
> We thank the reviewer for their detailed review and questions. We are pleased to read that they find our paper provides a solid theoretical basis for both routing and cascading and appreciate our significant performance gains. Below, we address the remaining questions and concerns.
>
> **Does cascade routing require extensive data to tune it effectively?**
> The hyperparameter optimization process for our proposed cascade routing method requires a minimal amount of training data. For routing algorithms, a single hyperparameter determines the trade-off between cost and quality, and this can often be set manually without extensive tuning. Cascading and cascade routing involve optimizing a slightly more complex set of parameters related to model selection in the cascade. However, we found that in cascade routing, assuming equal hyperparameters across stages already gives strong performance, significantly reducing the need for tuning.
>
> In contrast, thresholding-based cascade algorithms typically require explicit hyperparameter tuning to perform optimally. Despite this, the overall data requirement for all methods is limited: even a few hundred training samples provide sufficient signal to achieve robust parameter estimation for routing, cascading, and cascade routing. This ensures that our proposed method remains competitive and scalable in scenarios where training data availability is constrained. Furthermore, the data on which the hyperparameters are tuned do not have to be labeled, since it mainly depends on the quality and cost estimates rather than the actual qualities and costs.
>
> **Is the hyperparameter optimization scheme time-consuming and computationally expensive?**
> No, the optimization scheme is highly efficient. In our experiments, we parallelized the search for hyperparameters over 20 CPU cores, evaluating 1,000 configurations. This process completed in under 6 hours on a standard laptop. By assuming equal hyperparameters across models, manual tuning can replace automated optimization in some cases, further reducing the effort required. Compared to the computational cost of LLM inference, the resources needed for hyperparameter optimization are negligible.
>
> **How can we address very noisy quality and cost estimates especially for real-time or adaptive applications where these are challenging to maintain with high accuracy?**
> Cascade routing relies on accurate estimates to make optimal decisions, and noisy signals—particularly in quality estimates—can degrade its performance. However, we emphasize that cost estimates in most tasks, such as classification, are exact or very good. The main challenge, therefore, lies in obtaining robust and accurate quality estimates.
>
> Our current approach uses certainty-based metrics as proxies for quality, but these are inherently limited in high-noise scenarios. To address this, we propose two potential directions for improvement: (1) leveraging reward models specifically designed to predict output quality, and (2) integrating additional signals such as syntactic correctness or error metrics (e.g., runtime errors in code) to refine quality predictions. These strategies could enhance the system’s adaptability to noisy environments. However, initial experiments with the first technique we performed did not improve quality estimation. Therefore, further work on this aspect is required.
>
> Despite the noise in quality estimates, our experimental results demonstrate that cascade routing outperforms traditional routing and cascading frameworks under a wide range of conditions. For further notes on the significance of our 1-4% improvement, please refer to Q2 in our main reply.
>
> We hope that this addresses all the reviewers' questions and concerns. We are happy to discuss any further questions the reviewer might have.

---

> > ### Comment · Reviewer_pgV6 · 2024-11-25
> >
> > Thank you for your response. For now, I will maintain my score and look forward to further discussions between you and the other reviewers.

---

### Official Review · Reviewer_VrKy · 2024-11-02

**Soundness:** 3
**Presentation:** 3
**Contribution:** 2
**Rating:** 6
**Confidence:** 4

**Summary:**

Given the current landscape, where numerous fine-tuned models of varying sizes are tailored to specific tasks, this paper introduces a solution for selecting the most suitable model for a given user query, called Cascade Routing. This approach combines the adaptability of routing with the cost-efficiency of cascading, resulting in improved output quality while reducing computational costs, outperforming existing methods.
Furthermore, the paper offers a theoretical analysis of both existing routing and cascading methods, as well as the proposed Cascade Routing approach. Both theoretical analysis and experimental results consistently demonstrate the superiority of the proposed method over current alternatives.

**Strengths:**

1. This paper presents a theoretical analysis of existing LLM selection strategies and introduces a new cascading strategy that outperforms current methods under specific conditions.
2. The paper is generally well-articulated and easy to understand.

**Weaknesses:**

1. The paper lacks experimental results for other routing and cascade methods, such as [1] and the existing routing and cascading methods mentioned in the related work. These methods can also reduce the cost of LLMs without compromising response quality.
[1] Large Language Model Cascades with Mixture of Thought Representations for Cost-Efficient Reasoning

2. The strength of this paper's approach lies in achieving higher accuracy at the same computational cost, though it comes at the expense of longer runtime. Given that the accuracy improvement ranges are only between 1% to 4%, the authors need to demonstrate that the additional runtime is sufficiently minimal to justify the practicality of this approach.
Specifically, the authors should clearly state the runtime of different methods in the experiments presented in Figure 2 and Table 2 and include the runtime and AUC for the three methods: Linear Interpolation, Routing, and Cascade (Baseline) in Table 3.

3. The method proposed in this paper does not change the model order compared to the existing cascade approach, nor does it alter the methods for estimating cost and output quality. It simply expands the search space, resulting in limited contributions.

**Questions:**

1. As mentioned in the paper, the proposed method performs poorly when the quality and cost estimates are inaccurate, which raises concerns about its practical application.

2. I would like to know how much training data is required for the hyperparameter training of this method compared to existing routing and cascading approaches. I hope the authors can provide some comparative results.

---

> ### Author Response · Authors · 2024-11-21
>
> We thank the reviewer for their detailed review and questions. We are pleased to read that they find our paper well-written and appreciate our theoretical analysis. Below, we address the remaining questions and concerns and aim to clarify any outstanding points.
>
> **Why did you not include other routing and cascading methods?**
> The work mentioned by the reviewer is an instantiation of the baseline cascade with a different quality estimate. Existing cascading algorithms, as outlined in prior work, rely on thresholding mechanisms for quality estimates, which RouterBench does not directly support due to the pre-availability of answers. In our second experiment, we chose specific quality estimates from [Gupta et al., 2024], which are state-of-the-art for the given problem. While other estimates (e.g., from [1]) are viable, cascade routing can naturally integrate them, making it adaptable.
>
> **Could you include an analysis on runtimes, showing that the additional runtime required is sufficiently minimal?**
> Good question! We now present a new, detailed analysis of the expected latency of selection strategies in Appendix G and find that cascade routing slightly increases latency by 0.6 seconds compared to routing. This minor increase can be attributed to two factors: (1) smaller models incur low computational overhead, and (2) early stopping when a smaller model meets the required confidence level reduces latency. Moreover, we note that latency can be directly incorporated into our cost estimates to optimize a weighted combination of latency and monetary costs if the application values latency as well.
>
> **Can the authors provide the amount of training data required for hyperparameter optimization and compare it to the other methods?**
> The hyperparameter optimization process for our proposed cascade routing method requires a minimal amount of training data. For routing algorithms, a single hyperparameter determines the trade-off between cost and quality, and this can often be set manually without extensive tuning. Cascading and cascade routing involve optimizing a slightly more complex set of parameters related to model selection in the cascade. However, we found that in cascade routing, assuming equal hyperparameters across stages already gives strong performance, significantly reducing the need for tuning.
>
> In contrast, thresholding-based cascade algorithms typically require explicit hyperparameter tuning to perform optimally. Despite this, the overall data requirement for all methods is limited: even a few hundred training samples provide sufficient signal to achieve robust parameter estimation for routing, cascading, and cascade routing. This ensures that our proposed method remains competitive and scalable in scenarios where training data availability is constrained. Furthermore, the data on which the hyperparameters are tuned do not have to be labeled, since it mainly depends on the quality and cost estimates rather than the actual qualities and costs.
>
> **What are the contributions associated with the proposed method, given that it only expands the search space?**
> See Q1 of our main reply.
>
> **What are the practical applications of cascade routing given that current quality estimates are noisy?**
> Although quality estimates are inherently noisy, our algorithm demonstrates resilience to such inaccuracies. The consistent improvements observed, even under high-noise conditions, underline its practical utility. On a large scale, such incremental advancements translate to significant benefits in real-world applications, where even modest cost or accuracy improvements can have substantial impacts (see also Q2 of our main reply).
>
> We hope that this addresses all the reviewers' questions and concerns. We are happy to discuss any further questions the reviewer might have.

---

> > ### Comment · Reviewer_VrKy · 2024-12-03
> >
> > The author's response addressed most of my concerns. Although the improvement of the experimental results in this paper is very small (1-4%). In Q1 of the main reply, the authors mention that by proving the optimality of the proposed algorithm, they can be certain that there will be no (significant) further gains by proposing a new algorithm. I think this is an important contribution of this paper. Therefore, I would increase the score to 6.

---

### Official Review · Reviewer_fzmQ · 2024-11-02

**Soundness:** 1
**Presentation:** 3
**Contribution:** 1
**Rating:** 3
**Confidence:** 4

**Summary:**

This paper introduces cascade routing, a framework that unifies and generalizes two common LLM model selection strategies: routing (which selects one model per query) and cascading (which tries progressively larger models). The authors provide theoretical proofs for optimal strategies in both routing and cascading, then use these insights to develop cascade routing. Their approach allows dynamic model selection and sequencing based on both query difficulty and model specialization. Empirical results show 1-4% improvements over baselines across multiple benchmarks. The work's main contribution is providing a theoretical framework that unifies these previously separate approaches to model selection.

**Strengths:**

* The paper provides rigorous proofs for optimality conditions in both routing and cascading, establishing a solid mathematical framework for model selection.
* The paper combines two previously separate approaches (routing and cascading) into a single coherent framework, showing how they can complement each other.

**Weaknesses:**

* The paper shows only 1-4% accuracy improvements, which may not justify the significant complexity, computation and latency overhead of running multiple models sequentially. The practical value proposition needs stronger motivation.
* There is no discussion of end-to-end inference latency when running multiple models in sequence. While API costs are analyzed, total response time - a critical metric for real-world applications that affect users' experienc - is completely ignored.
* Simply using perplexity and uncertainty features to measure the models' response quality is not fair.
* Additionally, Chain-of-thought methods should be another baseline to compare with the cascading methods.

**Questions:**

* Given the significant latency overhead of running multiple models sequentially, could you provide a comprehensive analysis of end-to-end response times? How does this compare to simpler approaches like routing-only or ensemble methods?
* For quality estimation of model responses, have you considered more sophisticated evaluation methods beyond perplexity and uncertainty features? For example, using LLM-as-Judge to better evaluate the response's quality?
* The 1-4% accuracy improvement seems modest given the additional complexity and latency. Could you elaborate on specific use cases or scenarios where this trade-off would be clearly beneficial? Perhaps certain domains or applications where even small accuracy gains are highly valuable?

---

> ### Author Response · Authors · 2024-11-21
>
> We thank the reviewer for their detailed review and questions. Below, we address their remaining questions and concerns and aim to clarify any outstanding points.
>
> **How does latency compare to simpler approaches like routing-only or ensemble methods?**
> Good question! We now present a new, detailed analysis of the expected latency of selection strategies in Appendix G and find that cascade routing slightly increases latency by 0.6 seconds compared to routing. This minor increase can be attributed to two factors: (1) smaller models incur low computational overhead, and (2) early stopping when a smaller model meets the required confidence level reduces latency. Moreover, we note that latency can be directly incorporated into our cost estimates to optimize a weighted combination of latency and monetary costs if the application values latency as well.
>
> **For quality estimation of model responses, have you considered more sophisticated evaluation methods beyond perplexity and uncertainty features?**
> Yes, we have. However, a key requirement for cascading is the efficiency of response quality evaluation. Sophisticated methods such as LLM-as-a-Judge, while effective, are computationally expensive and would significantly diminish the utility of the cascade framework. Current estimates based on perplexity and uncertainty are informed by state-of-the-art methods in cascade quality estimation (Gupta et al. 2024).
>
> That said, we acknowledge the importance of exploring alternative evaluation strategies that balance cost and sophistication. Techniques such as lightweight heuristic-based scoring or smaller secondary models to approximate quality could serve as viable alternatives. For this purpose, we experimented with several classification reward models trained on user preferences to predict quality. However, we found that they do not provide a better signal for the quality estimator: their inclusion does not increase performance of the quality estimator. We therefore omitted them from the experiments.
>
> **Could you elaborate on specific use cases or scenarios where the trade-off between the additional complexity and the 1-4% accuracy improvements would be clearly beneficial?**
> We appreciate the reviewer’s concern about the modest improvement. While 1-4% might seem minor in isolation, the value of this gain is context-dependent and substantial in certain real-world scenarios:
> - **High-stakes decision-making**: Domains such as healthcare diagnostics, financial fraud detection, or legal document analysis often rely on models for large-scale classification or analysis. In these settings, small gains in accuracy directly translate to measurable improvements in outcomes. For instance, a 1% reduction in diagnostic error rates or false positives can have significant real-world impact, both ethically and economically. Further, these domains are often not latency-critical as they are run in batches.
> - **Large-scale operations**: In applications such as e-commerce recommendation systems, social media moderation, or news summarization, even fractional percentage gains in accuracy scale massively with large datasets. A 1% improvement in a system processing millions of requests daily could enhance user experience and engagement, while also reducing downstream errors. Once again, latency is not a key concern here.
>
> **Is the 1-4% accuracy improvement modest given the additional complexity and latency?**
> Please see Q2 of our main reply.
>
> **Should Chain-of-thought methods be another baseline to compare with the cascading methods?**
> We agree that chain-of-thought reasoning is a valuable technique. However, it is orthogonal to cascading algorithms, as both can be used in conjunction. For instance, our open-form experiment in Table 2 explicitly incorporates chain-of-thought reasoning to enhance response quality before cascading decisions are made.
>
> We hope that this addresses all the reviewers' questions and concerns. We are happy to discuss any further questions the reviewer might have.

---

> > ### Comment · Reviewer_fzmQ · 2024-11-24
> >
> > Thank you for your thoughtful rebuttal. However, it is still unclear to me what the latency metric reported in Appendix G is. I think a fair metric for latency measurement should be the TTFT (time-to-first-token) of your approach’s final response compared to the baseline’s final response. When you discuss your latency, it is unclear whether this refers to the cascading module’s execution latency or the whole end-to-end latency (TTFT). I think the latter is more important to the user’s experience. Therefore, I maintain my rating.

---

> > > ### Author Response · Authors · 2024-11-25
> > >
> > > We thank the reviewer for their comment. You are correct that we currently report the total time in Appendix G rather than TTFT. For classification tasks, these two metrics are equivalent, as there is only one output token. However, for open-ended generation, we acknowledge the reviewer's observation that TTFT is significantly higher for cascading approaches. As we mentioned in our earlier reply on specific use cases or scenarios, there are numerous applications where TTFT may not be a critical metric. Specifically, TTFT is primarily relevant for user-facing applications of LLMs, whereas it is less significant for large-scale analysis or classification tasks.
> > >
> > > Furthermore, we note that this limitation is not unique to cascade routing—it is inherent to all cascading approaches. Given the substantial body of work on cascading techniques (as discussed in Related Work), it is clear that researchers find the topic both interesting and applicable. Does the reviewer believe that this entire subfield would not warrant further exploration or acceptance based on the outlined limitation?

---

> > > > ### Comment · Reviewer_fzmQ · 2024-11-25
> > > >
> > > > I would like to clarify a few points:
> > > > 1. An LLM is inherently a generative model, not a discriminative one. Even when prompted to perform a one-token output classification task, the latency to generate the classification label for the final answer should still be defined as TTFT. Note that TTFT is defined as the time to the first token. I do not understand why the author claims that the classification task is significantly different from open-ended generation.
> > > > 2. I strongly disagree with the assertion that TTFT is not a critical metric. There are numerous algorithmic and system-level techniques aimed at optimizing different aspects to reduce computation overhead and latency. TTFT is generally an important metric that significantly affects user experience.
> > > > 3. I did not deny the usefulness of cascade routing in certain domains. However, if it leads to substantially increased latency, one should expect a corresponding significant improvement in response quality, such as what is achieved with test-time scaling methods like OpenAI’s o1 model. That said, achieving only a 1-4% quality improvement does not seem to justify such a large latency overhead.

---

> > > > > ### Author Response · Authors · 2024-11-25
> > > > >
> > > > > We thank the reviewer for their comment. We fully agree that TTFT is a critical metric in certain applications of LLMs, particularly in user-facing scenarios where tokens are streamed directly to the user. In such contexts, cascading-based approaches are suboptimal. However, there are numerous other applications of LLMs where tokens are not streamed, such as backend tasks involving large-scale analysis of user comments, news summarization, or content moderation. In these use cases, latency is either not a critical concern or total latency (i.e., the time until the full response is generated) is the more relevant metric. This is the metric we focus on in Appendix G.
> > > > >
> > > > > For classification tasks, we believe the situation differs because TTFT and total latency are effectively equivalent. This clarification was made because it explains the numbers presented in Appendix G, where TTFT for classification tasks aligns with the total latency.

---

### Official Review · Reviewer_nPNB · 2024-11-03

**Soundness:** 3
**Presentation:** 3
**Contribution:** 2
**Rating:** 6
**Confidence:** 4

**Summary:**

The paper explores the challenges of efficiently using large language models (LLMs) by evaluating existing model selection strategies: routing and cascading. Routing selects a single model per query, while cascading runs models sequentially until a satisfactory answer is found. However, routing lacks flexibility, and cascading is inefficient.

The paper contributes by deriving optimal strategies for both routing and cascading. Building on this analysis, the authors introduce "cascade routing," which combines the adaptability of routing with the cost-efficiency of cascading. Experiments demonstrate that cascade routing consistently outperforms existing methods across various scenarios, enhancing output quality and reducing computational costs. This approach offers a more unified and efficient solution to the model selection problem.

**Strengths:**

1. The paper frames routing and cascading as linear optimization problems, deriving optimal strategies for both. This provides a strong theoretical foundation lacking in prior work, which often relied on heuristics or simplifying assumptions. The theoretical framing allows for provable optimality under certain conditions.

2. Experimental evaluations on RouterBench consistently demonstrate improved performance, yielding a 1-4% increase in AUC across various noise levels and model set sizes, for both routing and cascading tasks.

3. Overall, the paper is well-written, clearly presented, and easy to follow, making it readily accessible to readers.

**Weaknesses:**

1. The core idea is an incremental approach that combines two existing methods, which somewhat limits the paper's novelty.

2. The effectiveness of cascade routing depends on precise quality and cost estimations. The paper acknowledges this limitation, demonstrating that performance gains decrease with less accurate estimates. Although various estimation methods are explored, further research is necessary to enhance their robustness in real-world scenarios.

**Questions:**

1. How does the computational cost of cascade routing, including the pruning strategy, scale with the number of available models? Have you tested this with significantly larger model sets, such as dozens or hundreds of models? Are there further optimizations that could enhance scalability?

2. The paper acknowledges the limitations of current quality estimation methods. Could you elaborate on potential future directions for improving these estimations? Are there specific features or model architectures that you find promising? Additionally, how does the choice of quality metric, such as accuracy versus user preference, impact the performance of cascade routing?

3. I suggest that the authors provide a more detailed error analysis of the results across different benchmarks. This analysis could offer insights into the types of queries where cascade routing excels or underperforms, guiding future improvements.

---

> ### Author Response · Authors · 2024-11-21
>
> We thank the reviewer for their detailed review and questions. We are pleased to read that they appreciate our strong theoretical foundation, assess our performance improvements to be consistent, and acknowledge that our paper is well-written. Below, we address their questions and concerns and aim to clarify all outstanding points.
>
> **How does the computational cost of cascade routing, including the pruning strategy, scale with the number of available models?**
> Yes, we have added new experiments in Appendix F, showing that "Slow" becomes significantly slower as more models are included. We also introduce a fifth variant, "Max-Depth," which limits the maximum depth of a cascade to three models. Notably, this approach improves speed without compromising the performance of cascade routing when the optimal depth is three models or fewer.
> Specifically, we evaluate five cascade routing variants for up to 80 models on random data. Our findings demonstrate that the "Slow" variant becomes infeasible beyond 20 models due to its high computational cost. Standard cascade routing can handle up to 40 models within a 1-second runtime, while "Max-Depth" further scales to 80 models within the same runtime. Furthermore, we now also observe a clear difference between the “No-Expect” and “Greedy” variants in terms of runtime. While Greedy remains computationally very cheap even for $80$ models, No-Expect has a significantly higher runtime, even obtaining higher runtimes than Max-Depth for $80$ models.
>
> **Could you elaborate on potential future directions for improving these estimations? Are there specific features or model architectures that you find promising?**
> Yes, as the reviewer correctly identifies, improving quality estimation is crucial to enhancing cascade routing's performance. In fact, due to our theoretical analysis, we can now be sure that any future improvements to model selection strategies are to be found in better quality estimation rather than in algorithmic improvements. While we experimented with reward models trained on user preferences, we found that these metrics correlated less with answer correctness than the mean log probabilities of generated tokens.
> Future work could explore hybrid approaches that combine token probabilities with contextual embeddings or fine-grained task-specific features. Additionally, using dynamic feedback loops to refine estimations based on live user interactions may further enhance robustness.
>
> **Could you provide a more detailed analysis of the results across different benchmarks?**
> We now include benchmark-specific results for the experiment in Section 5.2 in Appendix H. These results reveal that cascade routing’s relative improvement varies depending on the used model set. We could not find any specific structure in these improvements.
>
> **Does the paper provide any contribution beyond the introduction of cascade routing, which is a combination of two known methods?**
> See Q1 of our main reply.
>
> We hope that this addresses all the reviewers' questions and concerns. We are happy to discuss any further questions the reviewer might have.

---

> > ### Comment · Reviewer_nPNB · 2024-11-23
> >
> > Thank you for your thoughtful rebuttal. While I now have a deeper appreciation for the proposed paradigm's efficiency and potential future extensions, I still have reservations about its novelty. Therefore, I maintain my "weak accept" rating.

---

### Author Response · Authors · 2024-11-21

$\newcommand{R}{\textcolor{green}{tXch}}$
$\newcommand{S}{\textcolor{blue}{pgV6}}$
$\newcommand{T}{\textcolor{purple}{VrKy}}$
$\newcommand{U}{\textcolor{yellow}{fzmQ}}$
$\newcommand{V}{\textcolor{orange}{nPNB}}$

We thank the reviewers for their high-quality reviews and questions. We are particularly happy to read they find our paper well-written ($\V, \T, \R$), appreciate our solid theoretical basis ($\R,\S,\T,\U,\V$), and acknowledge our consistent performance gains ($\S,\V$). We identified two common questions between the reviewers and will also include a comprehensive overview of the changes made to the paper in response to the reviews.

**Q1. Does the paper provide any contribution beyond the introduction of cascade routing? ($\T,\V,\R$)**
Yes, while cascade routing unifies routing and cascading strategies, the paper’s contributions extend far beyond this specific algorithm. Here, we elaborate on the most significant contributions:
- **A Theoretical Framework with Rigorous Guarantees**: One of the key contributions of our paper is the derivation of optimal strategies for both routing and cascading. This approach offers provable guarantees for the performance of these strategies, addressing a fundamental gap in prior work, which relied on heuristics or empirical methods without rigorous foundations.
- **Cascade Routing**: Cascade routing is not just a straightforward combination of routing and cascading strategies; it is a theoretically informed algorithm explicitly designed to balance cost-efficiency and adaptability. Without this theoretical understanding of the problem, it would have been impossible to propose an algorithm that effectively combines both routing and cascading.
- **No Further Gains from Algorithmic Improvements:** A particularly impactful conclusion from our work is provided by our theoretical analysis: by proving the optimality of the proposed algorithms, we can be certain that no (significant) further gains can be made by proposing new algorithms. Therefore, the research community for routing and cascading can now focus on improving quality and cost estimation methods, rather than developing new routing and/or cascading algorithms.

**Q2. Is the 1-4% accuracy improvement modest given the additional complexity and latency? ($\U, \T, \R$)**
We argue that the reported 1-4% improvement is not modest when viewed in the broader context of the field. Several points support this:
- **Alignment with field benchmarks**: Improvements in routing and cascading algorithms have historically fallen within a similar range (Gupta et al. 2024, Ramirez et al. 2024, Jang et al. 2023, Nguyen et al. 2024). The consistency of our results across benchmarks shows that we improve upon prior work by as much as they improve over their own baselines, reaffirming the robustness of our contributions.
- **Practical implications of theoretical optimality**: Our results are not just empirical but grounded in theoretical guarantees. This ensures that any observed improvements are provably the best achievable under the current framework, making the method inherently more reliable and transferable to other applications.
- **Significance in specific domains**: In domains where the marginal value of improvement is high (e.g., regulatory compliance, critical automation systems), these consistent accuracy gains justify the additional complexity. Importantly, the consistency of our improvements ensures reliability.

**We made the following changes to our paper:**
- **Confidence Intervals**: We have added a discussion of confidence intervals in Appendix E. In Tables 1 and 2, we have underlined the numbers that are not statistically significantly lower than the highest value for clarity.
- **Further Runtime Analysis**: Additional experiments analyzing the runtimes of different cascade routing variants on random data are included in Appendix F. These experiments confirm that the "Slow" variant does not scale well, while the "Greedy" variant remains highly efficient for up to 80 models.
- **Latency**: The latencies associated with each method in Section 5.2 are analyzed in Appendix G. Our findings show that cascade routing is, on average, no more than 0.6 seconds slower than direct routing.
- **Benchmark-Specific Results**: Appendix H provides benchmark-specific AUC scores for the experiment discussed in Section 5.2.

---

### Author Response · Authors · 2024-12-04

We sincerely thank all five reviewers for their constructive comments and questions, which have significantly helped us improve our work. We are particularly encouraged by the reviewers' recognition of our paper's strengths, as summarized below:
$\newcommand{R}{\textcolor{green}{tXch}}$
$\newcommand{S}{\textcolor{blue}{pgV6}}$
$\newcommand{T}{\textcolor{purple}{VrKy}}$
$\newcommand{U}{\textcolor{black}{fzmQ}}$
$\newcommand{V}{\textcolor{orange}{nPNB}}$

**Solid Theoretical Basis**
- The presented framework provides a strong theoretical foundation to routing and cascading which was lacking prior work ($\V,\U,\T,\S,\R$).
- Cascade routing effectively combines routing and cascading for improved query handling ($\S,\U,\T$).

**Good Presentation**
- The paper is well written, clearly presented, and easy to understand ($\V,\T,\R$).

**Consistent Performance Gains**
- Our experimental evaluation shows that cascade routing demonstrates consistent performance improvements over routing and cascading ($\V,\S$).

We acknowledge that our initial submission had areas for improvement. We believe to have addressed the following main concerns by the reviewers:
- **Small Performance Gains**: We clarified that the gains were statistically significant, and that our theoretical optimality guarantees show that algorithmic improvements cannot significantly improve upon the presented method. This latter point was appreciated by Reviewer $\T$ who increased their score.
- **Novelty Cascade Routing**: We clarified that our novel reinterpretation of routing and cascading was essential in enabling the straightforward proposal and presentation of the cascade routing algorithm. Furthermore, the new theoretical foundation underlying routing and cascading was also required for the development of cascade routing.

In response to the reviewers’ thorough reviews, we have provided the following additional information and experiments which we incorporated in the revision of the paper:

- **Confidence Intervals**: We now include confidence intervals, indicating that our results are statistically significant.
- **Runtime Analysis**: We extended our ablation study, confirming our initial results regarding runtime improvements in the presented algorithm.
- **Latency**: We show that total latency is only increased slightly by cascade routing, and is generally lower than cascading.

We additionally clarified the following significant points to individual reviewers:
- **Applications**: While cascading approaches are inapplicable to user-oriented applications where time-to-first token is important, there are many applications that go beyond this use-case for LLMs, especially regarding high-stakes decision making and large-scale operations.
- **Hyperparameter Optimization is Cheap**: Hyperparameter optimization generally only takes a couple of hours on a couple of CPUs.

Once again, we deeply appreciate the valuable feedback and guidance provided by the reviewers.

Best regards,
The authors

---

### Meta-Review · Area_Chair_uX3j · 2024-12-20

**Metareview:**

The paper studies optimal strategies for routing and cascading in LLMs. Experiments demonstrate that the proposed combined approach outperforms routing or cascading when deployed alone.

Reviewers appreciated the importance of the questions asked, and applauded the theoretical contribution of formally posing this joint optimization problem. On the negative side, but routing and cascading are existing methods and, as such, blending the two was deemed somewhat incremental. The paper acknowledges that estimation of problem parameters may be difficult, which may further hinder the applicability of the method. Another recurring issue was the small comparative improvement over baselines: the optimization does not seem to yield significant dividends.

The authors addresses several concerns in the rebuttal phase, particularly adding experiments w.r.t. to scaling depth in routing and added more results on benchmarks; these are in the right direction, and more such results would strengthen the paper further. Latency results should be in TTFT, taking into account all possible overheads. The authors are encouraged to actually measure deployments. Performance of competitors should also be reported in a unified fashion.

**Additional Comments On Reviewer Discussion:**

Reviewers mostly remained concerned about (a) the small improvements, (b) the estimation of parameters needed for the problem, and (c) the correct way of measuring and reporting wall-clock latency.

---

### Decision · Program_Chairs · 2025-01-22

Reject